# Mannose metabolism inhibition sensitizes acute myeloid leukaemia cells to therapy by driving ferroptotic cell death

Keith Woodley[1], Laura S. Dillingh[2], George Giotopoulos [2,3], Pedro Madrigal [2,3,7], Kevin M. Rattigan[4], Céline Philippe [1], Vilma Dembitz [1], Aoife M. S. Magee[1], Ryan Asby[3], Louie N. van de Lagemaat [1], Christopher Mapperley [1], Sophie C. James[1], Jochen H. M. Prehn[5], Konstantinos Tzelepis [2,3,6], Kevin Rouault-Pierre [1], George S. Vassiliou [2,3], Kamil R. Kranc [1], G. Vignir Helgason [4], Brian J. P. Huntly [2,3] & Paolo Gallipoli [1] ✉

Resistance to standard and novel therapies remains the main obstacle to cure in acute myeloid leukaemia (AML) and is often driven by metabolic adaptations which are therapeutically actionable. Here we identify inhibition of mannose-6-phosphate isomerase (MPI), the first enzyme in the mannose metabolism pathway, as a sensitizer to both cytarabine and FLT3 inhibitors across multiple AML models. Mechanistically, we identify a connection between mannose metabolism and fatty acid metabolism, that is mediated via preferential activation of the ATF6 arm of the unfolded protein response (UPR). This in turn leads to cellular accumulation of polyunsaturated fatty acids, lipid peroxidation and ferroptotic cell death in AML cells. Our findings provide further support to the role of rewired metabolism in AML therapy resistance, unveil a connection between two apparently independent metabolic pathways and support further efforts to achieve eradication of therapy-resistant AML cells by sensitizing them to ferroptotic cell death.

Resistance to therapy leading to disease relapse is the most frequent cause of treatment failure in acute myeloid leukaemia (AML)[1] and commonly results from the emergence of genetic mutations, often within the therapeutic target[2,3]. However clinical and preclinical studies, in both solid[4] and haematological malignancies[5] have shown that early non-genetic adaptations might allow some cancer cells to withstand therapeutic stress, while allowing the development of a fully resistant phenotype through either established adaptive changes or the subsequent acquisition of genetic mutations[6]. Adaptive changes can be driven by metabolic rewiring and metabolism has emerged as a

therapeutically actionable vulnerability in AML[7], where specific metabolic adaptations arising as a result of driver mutations or in response to therapy have been reported[8–13].

D-Mannose is the 2-epimer of glucose and is a six-carbon sugar mostly used by cells for production of glycoconjugates rather than as an energy source. Mannose is transported into cells via transporters of the *SLC2A* group (GLUT) family and once within the cell is phosphorylated by hexokinase (HK) to produce mannose-6-phosphate (Man-6-P), which can mostly be either interconverted to the glycolytic intermediate fructose-6-phosphate (Fru-6-P) by mannose-6-phosphate isomerase

[1]Centre for Haemato-Oncology, Barts Cancer Institute, Queen Mary University of London, London, UK. [2]Wellcome - MRC Cambridge Stem Cell Institute, University of Cambridge, Cambridge, UK. [3]Department of Haematology, University of Cambridge, Cambridge, UK. [4]Wolfson Wohl Cancer Research Centre, Institute of Cancer Sciences, University of Glasgow, Glasgow, UK. [5]Department of Physiology & Medical Physics, Royal College of Surgeons in Ireland University of Medicine and Health Sciences, Dublin, Ireland. [6]Milner Therapeutics Institute, University of Cambridge, Cambridge, UK. [7]Present address: European Molecular Biology Laboratory, European Bioinformatics Institute, EMBL-EBI, Hinxton CB10 1SD, UK. ✉e-mail: p.gallipoli@qmul.ac.uk

(MPI) or directed into N-glycosylation via phosphomannomutase (PMM2)[14]. Since mannose plasma concentration is about 100-fold less than glucose concentration, the bidirectional MPI enzyme also plays a role in channelling glucose into the mannose metabolism (MM) pathway to feed the production of GDP-Mannose, a sugar donor for N-glycosylation reactions. Indeed, most of the mannose in N-glycans derives from glucose[15] and as a result MPI sits at a branching point between N-glycosylation and energy metabolism pathways, such as glycolysis and the hexosamine biosynthetic pathway (HBP) (Supplementary Fig. 1a). MPI plays an important role in embryonic development as *Mpi* knockout mice are embryonic lethal[16] and only hypomorphs rather than completely inactivating mutations are described in patients[17,18]. However partial loss of MPI function causes a congenital disorder of glycosylation (CDG; MPI-CDG) in humans which is successfully treated with mannose supplementation[19] while hypomorphic *Mpi* mice are viable[20]. Recent reports have highlighted a role for MPI in cancer cell proliferation or survival through varied mechanisms; regulation of TP53 O-linked-N-acetylglucosaminylation (O-GlcNAcylation) via modulation of HBP activity[21], reduction of cell surface receptor glycosylation and signalling[22] or modulating the susceptibility of several cancers to the toxic effects of high mannose diets[23]. Regardless of the mechanism, high MPI activity appears to provide a survival advantage in several cancer types. Interestingly, MPI was amongst the top drop-out genes in our published CRISPR-Cas9 screen aiming to identify sensitizers to the clinical grade FLT3-tyrosine kinase inhibitor (TKI) AC220 (quizartinib) in AML carrying activating FLT3 internal tandem duplication (ITD) mutations[9] and more recently was identified amongst 81 United States Food and Drug Administration (FDA) druggable genetic dependencies of a murine model of FLT3^ITD AML[24].

In this work, we therefore aim to test if inhibition of MPI and MM sensitizes AML cells to FLT3-TKI and standard chemotherapy and characterise the mechanistic consequences of MPI inhibition. We show that indeed MPI inhibition sensitizes AML cells to both cytarabine and FLT3-TKI therapy by priming them to ferroptotic cell death. Mechanistically we show this being secondary to the activation of the ATF6 arm of the unfolded protein response (UPR) which in turn impairs fatty acid metabolism in AML cells and leads to intracellular accumulation of polyunsaturated fatty acids (PUFA).

## Results

### High MPI expression correlates with worse prognosis in AML and a gene signature associated with enhanced oxidative phosphorylation and fatty acid metabolism

Given its high millimolar plasma concentration glucose generates the majority of cellular Man-6-P via MPI[15]. Our previous[9] liquid chromatography-mass spectrometry (LC/MS) experiments with uniformly-labelled ^13^Carbon(U-^13^C_6)-glucose in FLT3^ITD mutant cells treated with FLT3-TKI show that upon FLT3 inhibition, glucose metabolism is blocked at the level of Fru-6-P (Supplementary Fig. 1a and ref. 9.). This suggests that while glycolysis is inhibited, metabolic pathways branching from Fru-6-P, such as MM are still active and might play a cytoprotective role. Corroborating this notion, it is noticeable that other enzymes downstream of MPI in MM, i.e. GMPPB, were depleted in our published screen[9] (Supplementary Fig. 1a). Interestingly analysis of 2 published gene expression profiles of newly diagnosed AML cases associated with patient outcome shows that higher levels of *MPI* expression correlated with significant or borderline significant worse patient outcome (Fig. 1a and Supplementary Fig. 1b). *MPI* expression levels are higher in AML mononuclear cells (MNC) compared to normal bone marrow MNC (Fig.1b and Supplementary Fig. 1c-d) and particularly in FLT3^ITD compared to FLT3^WT AML (Fig.1c and Supplementary Fig. 1e) suggesting that MPI might have a prominent role in this AML subtype. When specifically focusing on FLT3^ITD mutant patients we observed a trend towards

improved survival in patients expressing low *MPI* levels (Supplementary Fig. 1f). However this was not significant possibly because of the small numbers. Gene-set enrichment analysis (GSEA) from several published gene expression datasets of AML samples at diagnosis highlighted oxidative phosphorylation, fatty acid oxidation and MYC targets gene signatures as being consistently upregulated in patients expressing high levels of *MPI* (Fig.1d and Supplementary Fig. 1g). Enhanced oxidative phosphorylation and fatty acid oxidation are known features of AML cells resistant to both current and novel therapies[10,13]. Further supporting the role of MPI in therapy resistance, analysis of paired diagnosis and relapsed samples following standard chemotherapy from 2 independent datasets[25,26] shows that *MPI* expression levels increase upon relapse (Fig.1e). Taken together these data suggest that MM and particularly *MPI* expression levels might play a role in resistance to both FLT3-TKI and standard therapies in AML. While this phenotype might be particularly associated with FLT3^ITD mutations, higher *MPI* expression also correlates with gene signatures associated with drug resistance in AML beyond the presence of FLT3^ITD mutations.

### MPI inhibition sensitizes both wild-type and FLT3^ITD mutant AML cells to novel targeted and standard therapies

To test the functional role of MPI in AML, we generated MPI KO and respective control non-targeting(NT)gRNA (hereafter Control) AML cells by CRISPR editing (Supplementary Fig. 2a) and confirmed that MPI KO sensitized FLT3^ITD AML cells to FLT3-TKI (Fig. 2a). Moreover, both FLT3^ITD and FLT3^WT MPI KO AML cells were sensitized to standard cytarabine (AraC) chemotherapy in competition assays (Fig. 2b). These effects were due to both reduced viability and proliferation and were rescued by adding mannose to the media thus confirming the specificity of MPI KO (Fig. 2c, d and Supplementary Fig. 2b-c). LC/MS analysis confirmed that mannose levels were reduced in MPI KO cells (Supplementary Fig. 2d). Similar phenotypic effects were observed using the MPI inhibitor MLS0315771 (hereafter MLS)[27] and following genetic silencing by shRNA (Fig.2e and Supplementary Fig. 2e–g). The specificity of MLS was confirmed by the lack of further toxicity in MPI KO cells (Supplementary Fig. 2h). Finally, in in vivo experiments, mice transplanted with MPI KO cells were more sensitive to AC220 and AraC resulting in both significant prolonged survival and reduction in leukaemia burden (Fig. 2f and Supplementary Fig. 2i–m). Taken together these results show that MPI plays a role in resistance to both FLT3-TKI and AraC in AML cells.

### MPI KO causes increased lipid uptake in AML cells

To understand how MPI inhibition increases sensitivity to AML therapies, we pursued an unbiased complementary metabolomic and transcriptional analysis (Supplementary Data 1 and 2). We focused on FLT3^ITD mutant cells treated with FLT3-TKI because MPI expression is higher in this AML subtype and adaptive resistance to FLT3-TKI is less well described thus necessitating a better understanding given the increasing clinical use of FLT3-TKI. Principal component analysis of both RNA-seq and untargeted metabolomics highlighted that MPI KO cells treated with FLT3-TKI separated from their respective Control and that the separation was partially rescued by addition of mannose to the media (Supplementary Fig. 3a, b). We identified over 600 metabolites across different pathways through untargeted metabolomics. Unexpectedly, the most striking change detected was the significant increase in the intracellular levels of both long-chain saturated fatty acids (SFA) and PUFA in MPI KO cells treated with FLT3-TKI which was partially rescued by mannose supplementation (Fig. 3a). Interestingly we also noted that both long and medium-chain acylcarnitines increase in AC220 treated MPI KO cells (Fig. 3b) and this was coupled with a reduction in tricarboxylic acid (TCA) cycle intermediates in the same conditions (Supplementary Fig. 3c). This metabolomic pattern

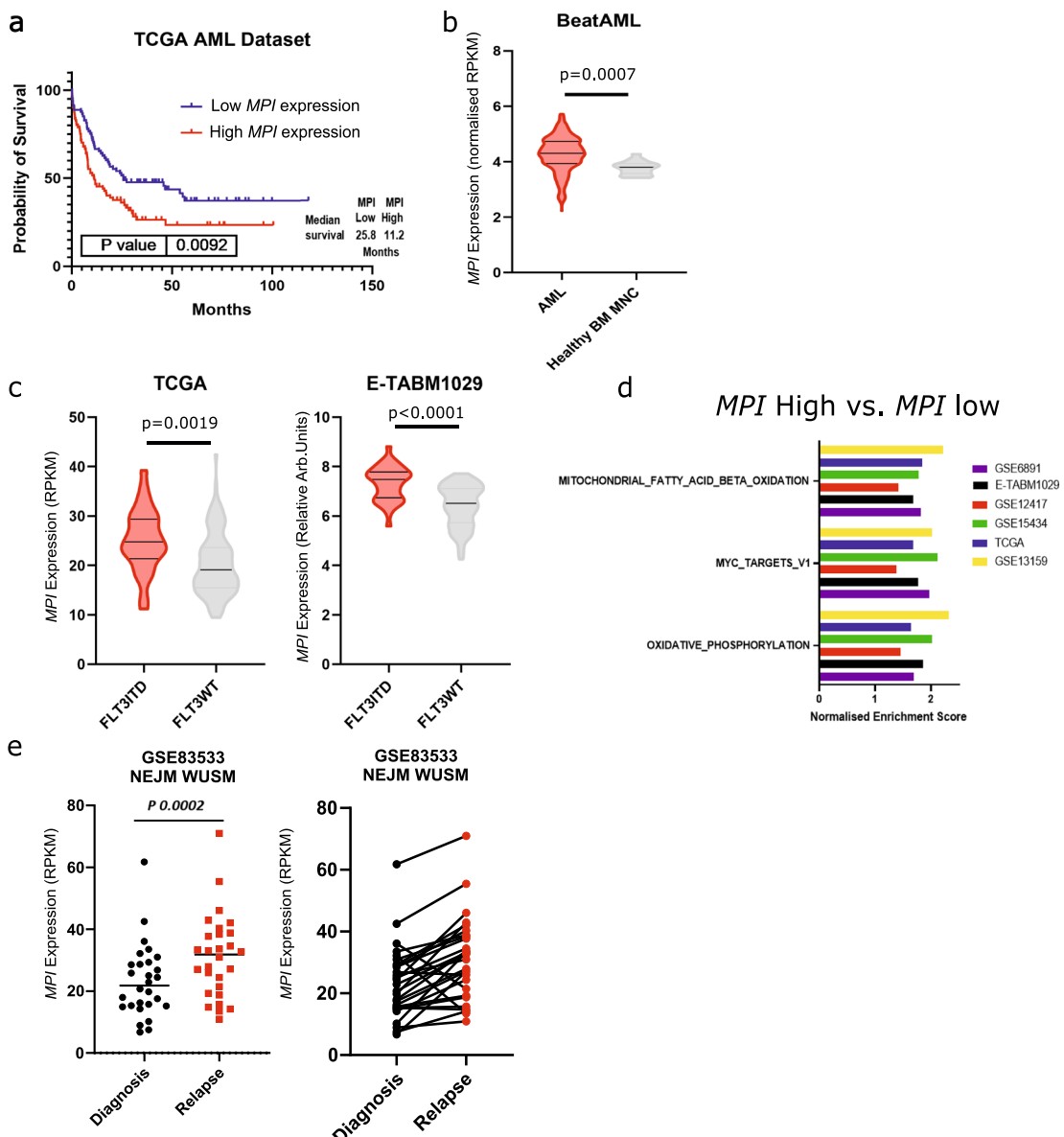

**Fig. 1 | High MPI expression correlates with worse prognosis in AML and a gene signature associated with enhanced oxidative phosphorylation and fatty acid metabolism. a** Kaplan-Meier curve comparing survival of patients from the TCGA AML cohort separated by the top 50% and bottom 50% of *MPI* expression (data obtained from https://www.cbioportal.org/). Log-rank (Mantel-Cox) test; (**b**) Violin plots of normalised *MPI* expression in AML samples compared to normal, healthy bone marrow mononuclear cells (MNC) from the BeatAML dataset (data obtained http://www.vizome.org/). Unpaired *t* test, two sided. *N* = 437 AML, *N* = 19 for healthy BM MNC; (**c**) Violin plots comparing *MPI* expression in FLT3$^{ITD}$ and FLT3$^{WT}$ AML samples from the TCGA dataset (left) and E-TABM1029 dataset (right). Unpaired *t* test, two sided. *N* = 28 FLT3$^{ITD}$ and *N* = 127 FLT3$^{WT}$ for TCGA, *N* = 38 FLT3$^{ITD}$ and *N* = 79 FLT3$^{WT}$ for E-TABM1029; (**d**) 3 significantly enriched gene signatures in *MPI* high expressing samples in 6 AML datasets (GSE6891, E-TABM1029, GSE12417, GSE15434, TCGA, and GSE13159); (**e**) Relative expression of *MPI* at diagnosis and relapse in paired samples from GSE83533 dataset and data from manuscript 10.1056/NEJMoa1808777 (NEJM WUSM), *N* = 28. Paired t-test, two sided. Source data are provided as a Source Data file. All data points presented. Violin plots presented as median and quartiles.

suggests both an increase in uptake of long-chain fatty acids and defective mitochondrial fatty acid oxidation (FAO) in treated MPI KO cells. To specifically test for fatty acid uptake, we labelled the cells with the saturated lipid probe C1-Bodipy 500/510 C12 and showed that MPI KO cells significantly increased fatty acid uptake (Fig. 3c and Supplementary Fig. 3d). To clarify if there was a preferential uptake of SFAs, monounsaturated fatty acids (MUFAs) or PUFAs, we used neutral Bodipy 493/503 to stain cells fed specifically palmitate (16:0 SFA), oleate (18:1 MUFA) and arachidonic acid (20:4 PUFA). While we observed significant increase in uptake of all subtypes of fatty acids in MPI KO cells, arachidonic acid uptake appeared to be particularly enhanced in the KO cells and was also rescued by mannose (Fig. 3d). This finding was corroborated by the metabolomic analysis as treated

MPI KO cells showed higher increase in their PUFA and PUFA-containing lipid species levels compared to MUFA and SFA (Supplementary Fig. 3e). Interestingly CD36, a fatty acid transporter with prognostic significance in AML[13,28], was transcriptionally upregulated in MPI KO cells (Supplementary Fig. 3f). To test if CD36 played a role in lipid uptake, we used its irreversible inhibitor sulfosuccinimidyl oleate (SSO)[29] and observed that, while CD36 inhibition reduced arachidonic acid uptake in MPI KO cells, it did not affect the uptake of the saturated lipid probe C1-Bodipy 500/510 C12 (Supplementary Fig. 3g-h). Together these data are consistent with MPI KO cells having an increased lipid uptake which, at least for PUFAs, is driven by increased CD36 expression. Interestingly CD36 is known to preferentially favour uptake of long chain PUFAs[30].

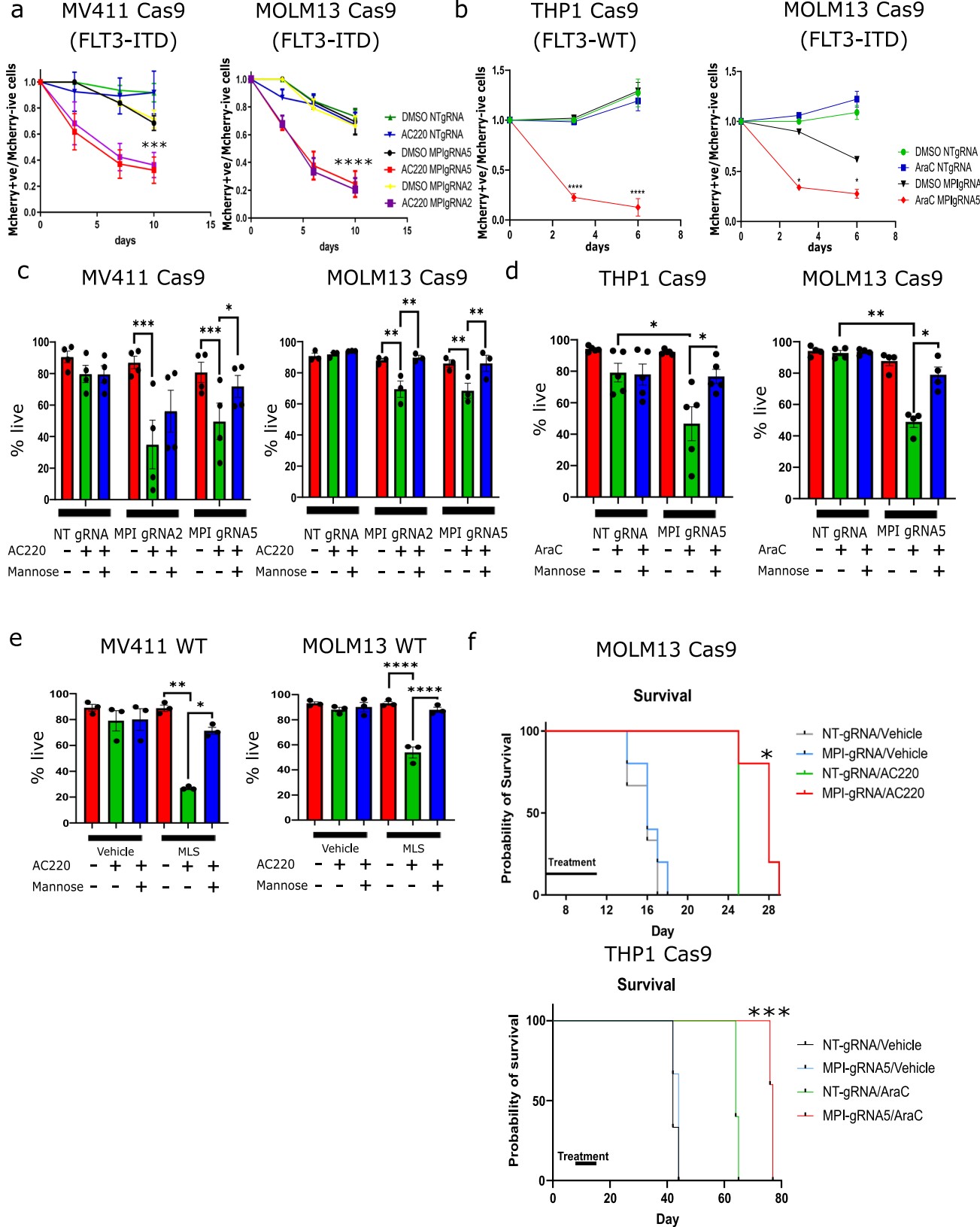

## MPI KO results in gene expression and metabolic changes consistent with reduced FAO in AML cells

We then turned our attention to FAO, given our metabolomic profile was suggestive of a defective FAO. GSEA analysis of RNA-seq data demonstrated that oxidative phosphorylation and fatty acid metabolism signatures were upregulated in Control compared to MPI KO cells both in the absence and, even more, in the presence of AC220. Similar genesets were enriched in AC220 treated MPI KO cells supplemented with mannose when compared to AC220 treated MPI KO cells (Fig. 4a and Supplementary Fig. 4a). Consistent with this, *MPI* expression levels were positively correlated with genes involved in fatty acid metabolism across both TCGA and BeatAML datasets (Supplementary Fig. 4b).

**Fig. 2 | MPI inhibition sensitizes both wild-type and FLT3ITD mutant AML cells to novel targeted and standard therapies. a** Competition growth assays measuring the ratio of mCherry positive to mCherry negative cells between WT MV411 and NT gRNA or MPI gRNA2 or gRNA5 (mCherry positive) with and without AC220 (1 nM) treatments (left) and WT MOLM13 (mCherry negative) and NT gRNA or MPI gRNA2 or gRNA5 (mCherry positive) with and without AC220 (1 nM) treatments (right) over 10 days. $N = 3$, 2 way Anova with Bonferroni correction for multiple comparisons; (**b**) Competition growth assays measuring the ratio of mCherry positive to mCherry negative cells between WT THP1 (mCherry negative) and NT gRNA or MPI gRNA5 (mCherry positive) with and without AraC (1 μM) treatments (left) and WT MOLM13 (mCherry negative) and NT gRNA or MPI gRNA5 (mCherry positive) with and without AraC (1 μM) treatments (right) over 6 days. $N = 3$, 2 way Anova with Bonferroni correction for multiple comparisons; (**c**) Percentage of live cells NT gRNA, MPI gRNA2 and MPI gRNA5 MV411 (left) and MOLM13 (right) cells treated with vehicle, AC220 (1 nM) or AC220 (1 nM) and mannose (100 μM), as indicated. Treated for 6 days, $N = 4$ for MV411, $N = 3$ for MOLM13, 1 way Anova with Tukey's correction for multiple comparisons; (**d**) Percentage of live cells of NT gRNA and MPI gRNA5 THP1 (left) and MOLM13 (right) cells treated with vehicle, AraC (1 μM) or AraC (1 μM) and mannose (100 μM). Treated for 3 days, $N = 4$ for MOLM13, $N = 5$ for THP1, 1 way Anova with Tukey's correction for multiple comparisons; (**e**) Percentage of live cells of WT MV411 (left) and WT MOLM13 (right) cells treated with vehicle, MLS0315771 (1 μM), AC220 (1 nM), mannose (100 μM) or combinations of these. Treated for 6 days $N = 3$, 1 way Anova with Tukey's correction for multiple comparisons; (**f**) Kaplan-Meier survival curve showing survival time of mice after transplantation with MOLM13 NT gRNA or MPI gRNA5 cells, with or without AC220 treatment (5 mg/kg, top) and THP1 NT gRNA or MPI gRNA5 cells, with or without cytarabine treatment (50 mg/kg, bottom). Log-rank (Mantel-Cox) test. For all panels, ns = not significant, *=$p < 0.05$, **=$p < 0.01$, ***=$p < 0.005$, ****=$p < 0.001$. Source data are provided as a Source Data file. All data presented as mean values ± SEM.

Moreover, RT-QPCR showed reduction in expression levels of *CPT1A* and *PPARA*, a master regulator of lipid catabolism, in MPI KO compared to Control cells (Supplementary Fig. 4c). Therefore, gene expression analysis suggest that treated MPI KO cells have defective FAO likely contributing to intracellular accumulation of both acylcarnitines and upstream fatty acids observed in metabolomic profiling.

To confirm reduction in FAO in MPI deficient cells, we grew cells in U-$^{13}$C$_{16}$-palmitate and found reduced palmitate labelling of several TCA intermediates in MPI KO cells, a phenotype rescued by mannose (Fig. 4b and Supplementary Fig. 4d). Real-time metabolic flux analysis also showed that MPI KO cells had reduced oxygen consumption rate (OCR) and ATP-linked OCR compared to Control, with the OCR further reduced in the presence of AC220 and rescued in the presence of mannose (Fig. 4c). Similar findings were observed in THP1 cells treated with AraC (Supplementary Fig. 4f). To specifically assess FAO we measured OCR using substrate limiting media in the presence of only palmitate. We first observed that with palmitate as the sole substrate, mannose supplementation rescued the respiration defect in treated MPI KO cells suggesting that MPI KO cells can engage efficiently in FAO only when supplemented with mannose (Supplementary Fig. 4e). Moreover, in substrate limiting media with palmitate, both basal respiration and ATP production were significantly reduced in KO cells compared to Control following AC220 treatment with the effects rescued by mannose (Fig. 4d). These data suggest MPI KO cells do not use palmitate efficiently as a source of energy, particularly in the presence of AC220, with this phenotype reversed by mannose. Indeed adding the CPT1A and FAO inhibitor etomoxir[11] to AC220 or AraC did not further reduce basal respiration or ATP production in MPI KO cells. However, the re-addition of mannose resensitized MPI KO cells to the effects of etomoxir on OCR (Fig. 4e and Supplementary Fig. 4g). These effects correlated with changes in viability in response to etomoxir in MPI KO cells, as no reduction in viability beyond the effects of AC220 was observed in response to etomoxir, unless cells were cultured in the presence of mannose (Fig. 4f). Conversely viability of AC220 or AraC treated MPI KO was rescued by the PPARA agonist and FAO activator fenofibrate[31] to a degree similar to mannose (Supplementary Fig. 4h). Finally we did not rescue viability of treated MPI KO cells using the cell permeable TCA cycle intermediate dimethyl-alpha-ketoglutarate (DMaKG) (Supplementary Fig. 4i). These data suggest that MPI KO cells have defective FAO but the effects on viability of MPI KO are driven by both defective FAO and enhanced lipid uptake causing an increase in intracellular fatty acid levels rather than simply a reduction in TCA cycle activity and oxidative phosphorylation.

## MPI KO cells activate the ATF6 arm of the UPR to inhibit FAO in AML cells

MPI deficiency leads to a defect in protein glycosylation[17] and as predicted MPI KO cells showed reduced protein glycosylation as confirmed by their lower lectin staining. This was also associated with concurrent accumulation of misfolded protein (Supplementary Fig. 5a, b). These effects correlated, as expected, with upregulation of UPR genes in MPI KO cells although in particular the ATF6 arm of the UPR, which is particularly sensitive to changes in glycosylation[32], was upregulated (Fig. 5a). To validate this, we transduced cells with an ATF6-YFP reporter lentivirus (Supplementary Fig. 5c)[33] and observed significant ATF6 activation in MPI KO cells which was reversed by mannose (Fig. 5b). We then used immunofluorescence to track localisation of ATF6 in MPI KO cells and showed that ATF6 localised in the nucleus of MPI KO cells and this was reversed by mannose (Fig. 5c, d and Supplementary Fig. 5d). Similarly western blot showed higher levels of cleaved ATF6 consistent with its activation in MPI KO cells compared to Control cells which was rescued by mannose (Supplementary Fig. 5e). Interestingly both in the ATF6 reporter assay and by western blot, AC220 treatment appeared also to reduce ATF6 activation although this was not seen by immunofluorescence. This discrepancy might reflect sensitivity of the assay and/or be related to the effects of AC220 on protein translation/ER load and ATF6 activation at the assessed timepoint. Conversely we did not observe consistent significant changes in the activation of the other branches of the UPR in MPI KO cells (Supplementary Fig. 5f) To functionally test the effects of ATF6 activation in MPI KO cells, we used the highly specific ATF6 inhibitor Ceapin A7 which retains ATF6 in the endoplasmic reticulum thus preventing its nuclear translocation and ability to activate transcription of its target genes[34,35]. Treatment with Ceapin A7 phenocopied mannose supplementation by reversing ATF6 cleavage and nuclear localisation (Fig. 5c, d and Supplementary Fig. 6a). Ceapin A7 also rescued viability and OCR of MPI KO cells treated with AC220 to the same degree of mannose and resensitized MPI KO cells to the effects of etomoxir on OCR (Fig. 5e, f). Similar effects were observed following silencing of ATF6 (Supplementary Fig. 6b, c). Conversely both the UPR activator with reported preferential activity on ATF6 AA147[36] and the canonical glycosylation inhibitor and UPR activator tunicamycin mimicked the effects of MPI KO on cell viability and prevented mannose rescue (Supplementary Fig. 6d, e). Finally inhibitors of the other UPR branches did not affect viability of treated MPI KO cells (Supplementary Fig. 6f). Therefore despite the fact that the UPR response is often non-selective and there is evidence that ATF6 can lead to activation of other arms of the UPR response[37,38], we observe in our system a preferential activation of ATF6. This might be due to low level and chronic endoplasmic reticulum stress activating preferentially the ATF6 arm of the UPR. Interestingly this is consistent with previous literature showing that ATF6 is able to refold proteins and prevent further stress and activation of the other arms of UPR which happens only when the refolding ability driven by ATF6 is overwhelmed[37,39]. Activation of the ATF6 arm of the UPR has been reported to inhibit the transcription/activity of the master regulator of lipid catabolism PPARA[40] and interestingly both Ceapin A7 and AA147, beside modulating the expression of canonical ATF6 targets *ERO1B*

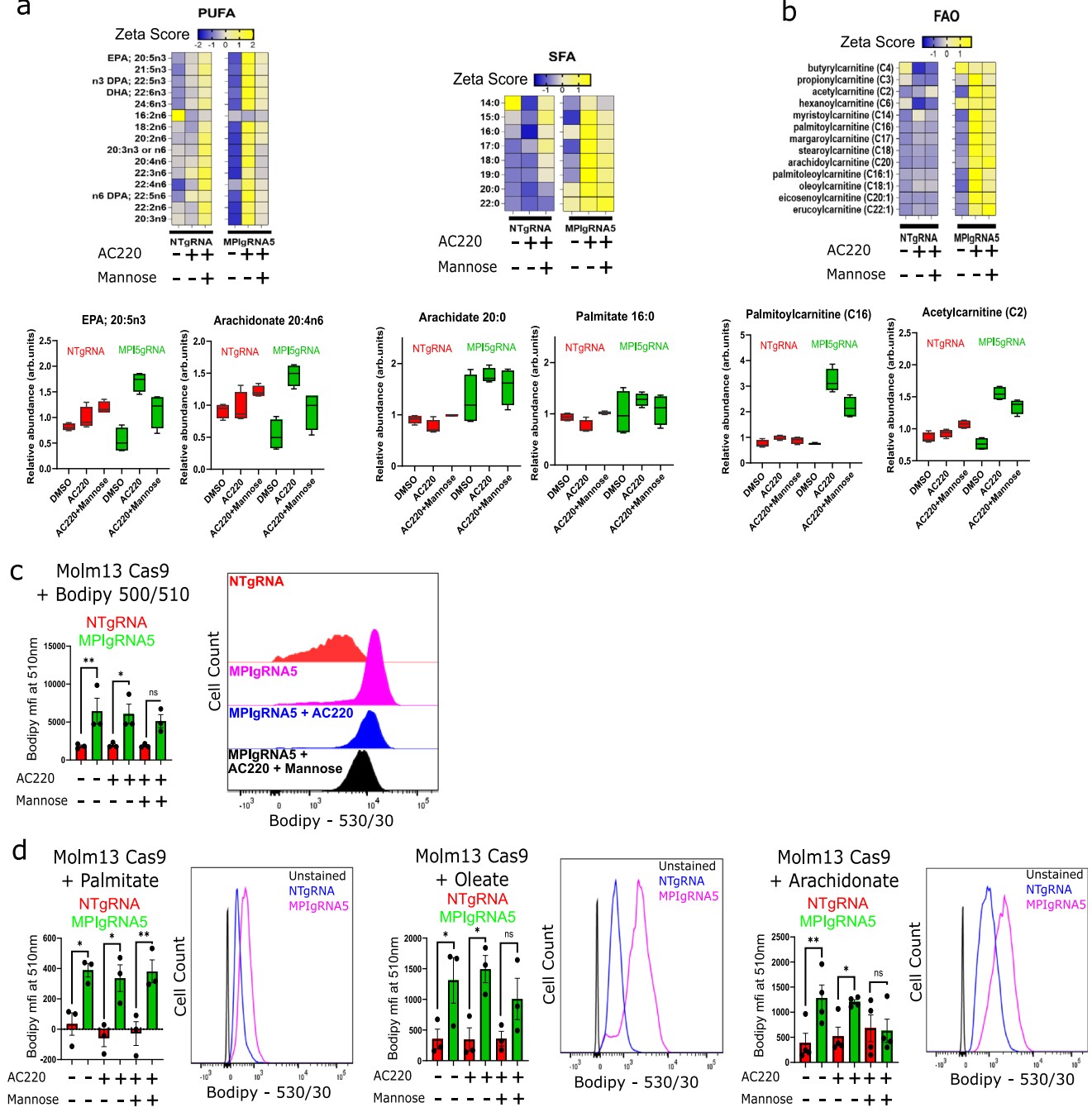

**Fig. 3 | MPI KO causes increased lipid uptake in AML cells. a** Heat maps of Z-scores for relative changes of long chain polyunsaturated fatty acids (PUFA - n3, n6 and n9, left) and long chain saturated fatty acids (SFA, right), with box plots of selected lipids of each type below the heat maps from global metabolomics of MOLM13 NT gRNA or MPI gRNA5 cells treated with vehicle, AC220 (1 nM) or AC220 (1 nM) and mannose (100 μM) as indicated for 48 h, *N* = 4; (**b**) Heat maps of Z-scores for relative changes of fatty acid oxidation metabolism related fatty acid species, with box plots of selected lipids below the heat maps from global metabolomics of MOLM13 NT gRNA or MPI gRNA5 cells treated with vehicle, AC220 (1 nM) or AC220 (1 nM) and mannose (100 μM) as indicated for 48 h, *N* = 4; (**c**) Uptake of fluorescent C1-Bodipy 500/510 C12 by NT gRNA or MPI gRNA5 MOLM13 cells treated with vehicle, AC220 (1 nM) or AC220 (1 nM) and mannose (100 μM) for 24 h, with a representative flow cytometry plot (right). *N* = 3, 1 way Anova with Tukey's correction for multiple comparisons; (**d**) Uptake of palmitate (50 μM, left), oleate (50 μM, middle) and arachidonic acid (1 μM, right) by NT gRNA or MPI gRNA5 MOLM13 cells treated with vehicle, AC220 (1 nM) or AC220 (1 nM) and mannose (100 μM) for 24 h as indicated and measured by Bodipy 493/503 neutral lipid stain with representative flow cytometry plots comparing uptake by NT gRNA and MPI gRNA5 cells (with unstained sample, to the right of each graph). *N* = 3 for palmitate and oleate, *N* = 4 for arachidonic acid, 1 way Anova with Tukey's correction for multiple comparisons. For all panels, ns = not significant, \*=*p* < 0.05, \*\*=*p* < 0.01, \*\*\*=*p* < 0.005, \*\*\*\*=*p* < 0.001. Source data are provided as a Source Data file. All data presented as mean values ± SEM, box plots presented as median with upper and lower quartiles as bounds of box and whiskers as max and min of distribution.

and *HERPUD1* as expected, respectively increased and decreased the expression of *PPARA* and *CPT1A*, the rate limiting step in FAO, thus phenocopying the transcriptional consequences of MPI KO and mannose rescue (Supplementary Fig. 6g, h). Interestingly, *MPI* expression levels were negatively correlated with *ERO1B* and *HERPUD1* expression across both TCGA and BeatAML datasets (Supplementary Fig. 6i). Taken together these data support a model where MPI KO cells preferentially activate the ATF6 arm of the UPR leading to reduced FAO.

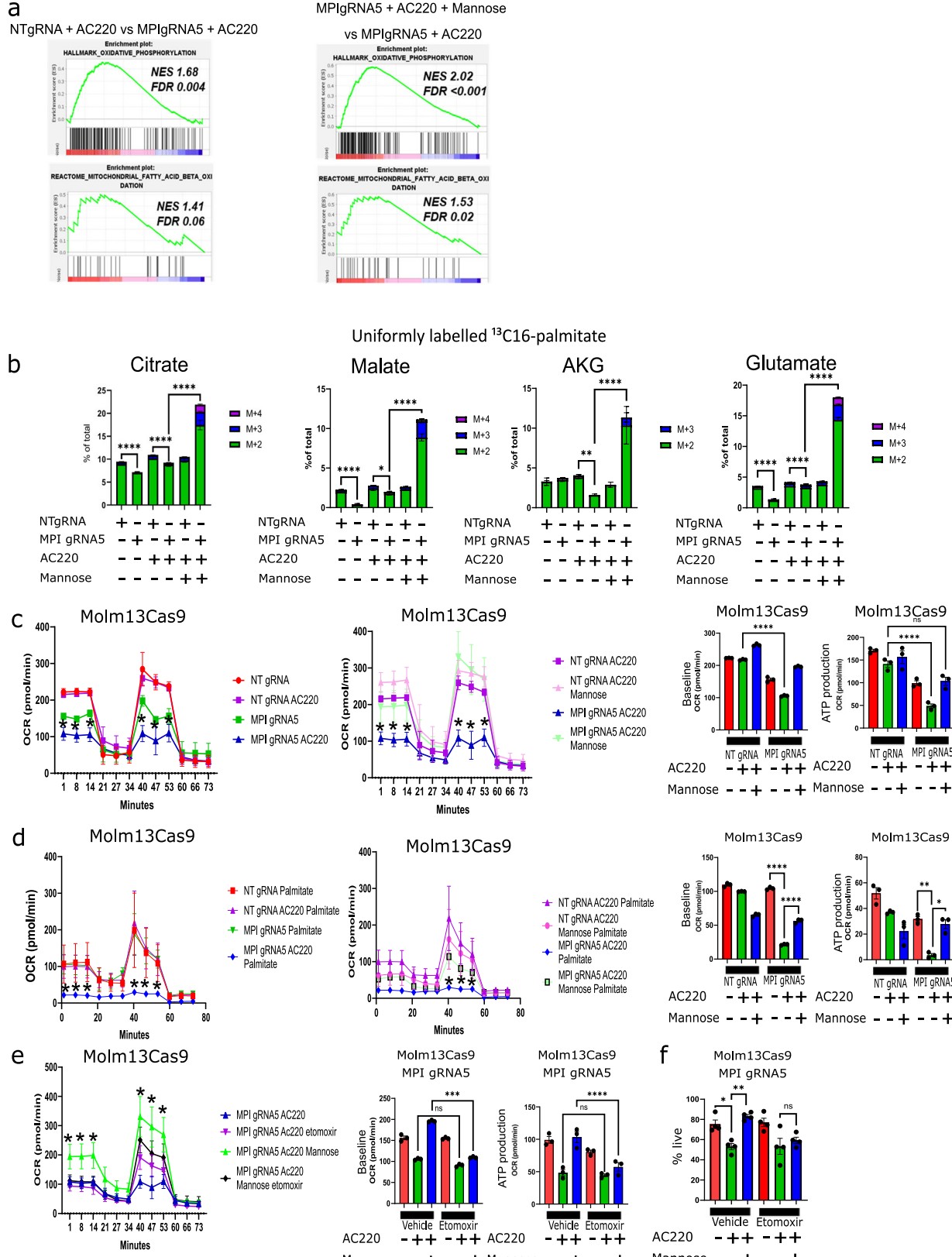

## MPI KO drives lipid peroxidation and ferroptotic cell death in AML cells

The metabolic phenotype of MPI KO cells with accumulation of PUFA and reduced FAO is reminiscent of that observed in cancer cells prone to ferroptotic cell death, i.e. clear cell renal carcinoma[41,42]. Moreover MPI KO cells had lower levels of intracellular cysteine and higher levels

of 4-hydroxy-nonenal-glutathione and ophthalmate (Fig. 6a) markers of oxidative stress and lipid peroxidation[43,44] which was further increased by AC220 a known driver of oxidative stress in these cells[9]. Interestingly a ferroptotic gene signature[45] was upregulated in treated MPI KO cells compared to both Control and MPI KO mannose treated cells (Fig. 6b). These observations led us to test if cell death of treated MPI KO cells was

**Fig. 4 | MPI KO results in gene expression and metabolic changes consistent with reduced FAO in AML cells. a** GSEA for Oxidative phosphorylation and Fatty acid oxidation signatures from RNA sequencing data comparing MOLM13 NTgRNA and MPIgRNA5 treated with AC220 (left) and MOLM13 MPIgRNA5 treated with AC220 or AC220 and mannose (right), FDR and NES from 1000 permutations; (**b**) Percentage of TCA cycle intermediates and associated metabolites labelled with $^{13}$C from $^{13}C_{16}$-palmitate in MPI KO and NT MOLM13 cells treated with AC220 and mannose as indicated for 24 h along with 50 μM $^{13}C_{16}$-palmitate. $N = 5$, 1-way anova with Tukey's correction; (**c**) SEAHORSE MitoStress tests showing oxygen consumption rate (OCR) of NTgRNA and MPIgRNA5 MOLM13 cells treated with vehicle, AC220 or AC220 and mannose as indicated after 72 h of treatment, $N = 3$, 2-way Anova with Sidak's correction (two left panels). Baseline OCR and ATP production of NTgRNA and MPIgRNA5 MOLM13 cells treated with vehicle, AC220 or AC220 and mannose as indicated after 72 h of treatment, $N = 3$, 1-way Anova with Tukey's correction (2 right panels); (**d**) SEAHORSE MitoStress tests showing OCR comparing MPIgRNA5 cells cultured overnight in substrate limited RPMI media without FBS and glutamine treated with vehicle, AC220, mannose or palmitate (50 μM) as indicated. $N = 3$, 2-way Anova with Sidak's correction (two left panels). Baseline OCR and ATP production of NTgRNA and MPIgRNA5 cells treated with palmitate (50 μM) and vehicle, AC220 or mannose as indicated, $N = 3$, 1-way Anova with Tukey's correction (2 right panels); (**e**) SEAHORSE MitoStress tests showing OCR comparing MPIgRNA5 cells treated with AC220, mannose after 72 h in combinations as indicated with or without etomoxir (50 μM), $N = 3$, 2-way Anova with Sidak's correction (left panel). Baseline OCR and ATP production of MPI gRNA5 cells treated with vehicle, AC220, mannose after 72 h in combinations as indicated with or without etomoxir (50 μM). $N = 3$, 1-way Anova with Tukey's correction (2 right panels); (**f**) Percentage of live MPIgRNA5 MOLM13 cells treated with vehicle, etomoxir, AC220, mannose or in combinations 6 days after treatment. $N = 4$, 1-way Anova with Tukey's correction. For all panels, ns = not significant, *=$p < 0.05$, **=$p < 0.01$, ***=$p < 0.005$, ****=$p < 0.001$, data presented as mean values ± SEM unless stated.

driven by lipid peroxidation and ferroptosis. Indeed Bodipy 581/591 C11 staining of MPI KO and knock-down (KD) cells confirmed higher levels of lipid peroxidation in treated MPI KO/KD cells, that was rescued by mannose (Fig. 6c and Supplementary Fig. 7a, b). Moreover, cell death and induction of lipid peroxidation in both AC220 and AraC treated MPI KO cells was rescued by the radical-trapping antioxidant ferrostatin and liproxstatin, known anti-ferroptosis agents[46] (Fig. 6d and Supplementary Data Fig. 7c–e). Interestingly Ceapin A7 and AA147 respectively mimicked and inhibited the effects of mannose on lipid peroxidation supporting the role of ATF6 activation in driving the ferroptotic phenotype (Supplementary Fig. 7f, g). Conversely neither PERK or IRE1 inhibition had significant effects on lipid peroxidation (Supplementary Fig. 7h, i). MPI KO cells were more sensitive to PUFA (arachidonic acid) toxicity which is rescued by mannose and SSO thus reinforcing the role of CD36 in PUFAs uptake (Supplementary Fig. 8a, b). Conversely growing MPI KO cells in delipidated media reduced their sensitivity to AC220 (Supplementary Fig. 8c). However SSO did not affect lipid peroxidation (Supplementary Fig. 8d) probably because it affects uptake of only some fatty acid species (Supplementary Fig. 3g, h) and has no effects on FAO. This also suggests that sensitivity to arachidonic acid toxicity is not just due to increased lipid peroxidation. MPI KO cells were also more sensitive to the ferroptotic inducing agents erastin and RSL3 (Supplementary Fig. 8e). Interestingly, and consistent with the reduced cysteine levels in MPI KO cells, surface expression of the cysteine transporter and erastin target SLC7A11/xCT was downregulated in MPI KO cells further explaining their tendency to undergo ferroptosis (Supplementary Fig. 8f). We also did not detect significant upregulation of cleaved PARP in treated MPI KO cells and neither the apoptosis/pan caspase inhibitor Z-VAD or necroptosis inhibitor Necrostatin-1 rescued cell death in both AC220 and AraC treated MPI KO cells (Supplementary Fig. 8g, h). Interestingly Necrostatin-1 has been shown to also rescue ferroptosis in other cellular models[47] but we did not observe that and therefore we conclude necroptosis is not driving cell death in our system. Finally to validate the role of ferroptosis induction in driving cell death in vivo mice transplanted with THP1 Control and MPI KO cells were treated with AraC ± liproxstatin. As expected liproxstatin treatment reduced disease latency in AraC treated mice transplanted with MPI KO cells and this correlated with reduced staining of the leukaemic cells for 4-HNE, a marker of lipid peroxidation, in vivo (Fig. 6e, f and Supplementary Fig. 8i–j). Overall these data show that, following chemo/targeted therapy treatment the metabolic rewiring caused by MPI KO primes the cells to lipid peroxidation and ferroptotic cell death.

## MPI depletion in primary FLT3$^{ITD}$ AML samples causes ATF6 activation, lipid peroxidation and sensitization to FLT3-TKI therapy

We finally validated the effects of MPI depletion in primary FLT3$^{ITD}$ AML samples (Supplementary Table 1). Following MPI KD (Supplementary Fig. 9a, b), FLT3$^{ITD}$ primary AML MNC were sensitized to FLT3-TKI therapy and this could be rescued by mannose (Fig. 7a). Conversely normal CD34$^+$ cells were not sensitive to MPI KD both in the absence or presence of AC220 (Fig. 7b). Consistent with observations in AML cell lines, MPI KD primed FLT3$^{ITD}$ AML MNC to lipid peroxidation following FLT3-TKI therapy and led to higher levels of nuclear ATF6, with both effects rescued by the addition of mannose (Fig. 7c, d). Conversely a smaller induction of lipid peroxidation was observed in normal CD34$^+$ cells upon MPI KD which was not enhanced by FLT3-TKI or rescued by mannose (Supplementary Fig. 9c). Moreover the viability of primary MPI KD FLT3$^{ITD}$ AML MNC treated with AC220 was rescued by ferrostatin and liproxstatin confirming that the mechanism of death in primary AML MNC was also due to ferroptosis (Fig. 7e). These experiments validate the phenotype driven by MPI inhibition in primary AML MNC while demonstrating that MPI KD is tolerated by normal hematopoietic stem/progenitor cells (HSPC).

## Discussion

Our data show that targeting MPI and MM sensitizes AML cells to AraC and FLT3-TKI. Recent reports, including one in leukaemic cells[48], suggest that MPI could be a therapeutic target in cancer therapy[22,23]. Most of these reports however suggest that MPI inhibition would limit the ability of cancer cells to withstand the toxicity of an exogenous high-dose mannose diet. Our data instead support the role of MPI, at physiological concentrations of mannose, as an enzyme important in supporting AML cells survival following therapy through its ability to modulate MM, protein glycosylation and activity of the ATF6 arm of the UPR. It is worth noting that the effects of MPI inhibition in vivo could be blunted by the endogenous mannose production and dietary mannose and this could partly limit the role of MPI as a therapeutic target. Conversely, given that glucose, which is 100-fold more abundant than mannose in human plasma[14], is the biggest source of mannose within the cells[15] and considering the higher expression levels of MPI in AML samples, it is plausible that MPI inhibition will be detrimental to AML cells survival and its effects not rescued sufficiently by the relatively low levels of mannose in plasma. This outcome would be expected particularly if associated with reduced glucose metabolism, such as when combined with FLT3-TKI. Interestingly congenital MPI defects in humans are successfully treated with mannose supplements suggesting that physiological levels of mannose are unable to overcome glycosylation defects in MPI deficient patients.

An ideal drug target requires a robust therapeutic window in order to be exploitable. Although the embryonic lethality of *Mpi* in animal models is a reason for concern, our preclinical data suggest that HSPC are less reliant on MPI activity than AML cells. Moreover the lack of hematopoietic defects in humans carrying MPI mutation (albeit in a hypomorphic state) suggest that MPI inhibition might be tolerable for

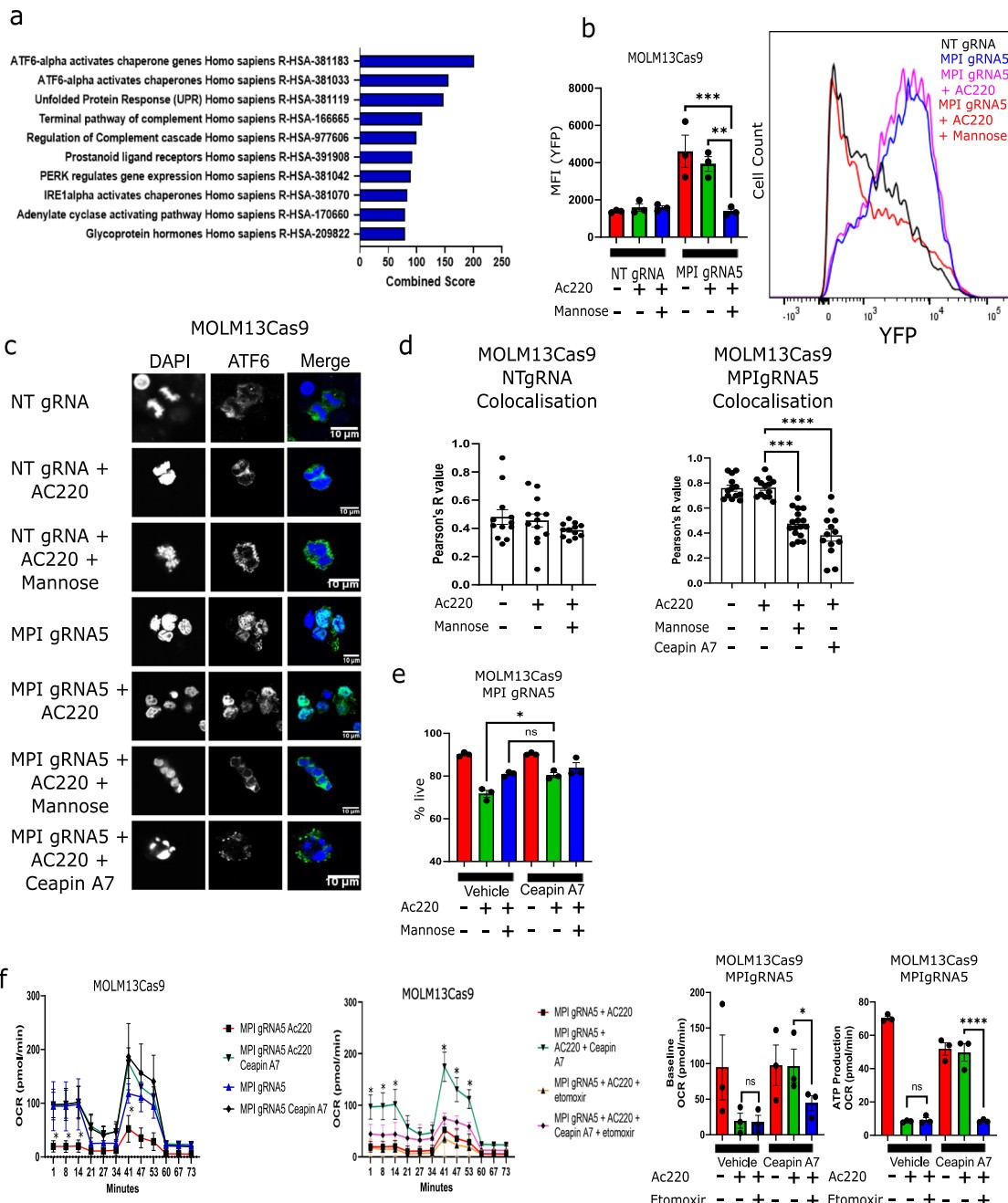

**Fig. 5 | MPI KO cells activate the ATF6 arm of the unfolded protein response (UPR) to inhibit oxidative phosphorylation metabolism in AML cells. a** Top combined score gene signatures from RNA sequencing enriched in MPI gRNA5 MOLM13 cells compared to NT gRNA MOLM13 cells. Analysis performed with Enrichr; **b** Mean fluorescence intensity of YFP from MOLM13 NT gRNA or MPI gRNA5 cells expressing a YFP linked ATF6 reporter treated with vehicle, AC220 (1 nM), mannose (100 μM) or combinations as indicated (left panel) with a representative flow cytometry plot (left right panel). $N = 3$, 1 way Anova with Tukey's correction for multiple comparisons; (**c**) Confocal microscopy images of NT gRNA and MPI gRNA5 MOLM13 cells stained for DAPI (blue, 1st column) and ATF6 (green, 2nd column) with a merged image (3rd column), treated with vehicle, AC220 (1 nM), mannose (100 μM), Ceapin A7 (1 μM) or in combinations as indicated 72 h after treatment. Experiment performed 3 times independently; (**d**) Colocalisation analysis from immunofluorescence images of NTgRNA (left) and MPIgRNA5 (right) MOLM13 cells treated with vehicle, AC220 (1 nM), mannose (100 μM) and Ceapin A7 (1 μM) in combinations as indicated. Analysis performed with Coloc2 plugin in

ImageJ, ordinary 1 way Anova with Tukey's correction for multiple comparisons, $N = 13$ for NTgRNA, $N = 14$ for MPIgRNA5; (**e**) Percentage of live cells of MPI gRNA5 MOLM13 cells treated with vehicle, AC220 (1 nM), mannose (100 μM), Ceapin A7 (1 μM) or in combinations as indicated 72 h after treatment. $N = 3$, 1 way Anova with Tukey's correction for multiple comparisons; (**f**) SEAHORSE MitoStress assay comparison of MPI gRNA5 MOLM13 cells, treated with AC220 (1 nM), Ceapin A7 (1 μM), etomoxir (10 μM) or in combinations as indicated 72 h after treatment. $N = 3$, 2 way Anova with Sidak's correction for multiple comparisons, with MPI gRNA5 AC220 as control population (2 left panels). Baseline OCR and ATP production of NT gRNA and MPI gRNA5 cells treated with vehicle, AC220 (1 nM), Ceapin A7 (1 μM), etomoxir (50 μM) or in combinations as indicated, $N = 3$, ns = not significant, 1 way Anova with Tukey's correction for multiple comparisons (2 right panels). For all panels, ns = not significant, *=$p < 0.05$, **=$p < 0.01$, ***=$p < 0.005$, ****=$p < 0.001$. Source data are provided as a Source Data file. All data presented as mean values ± SEM.

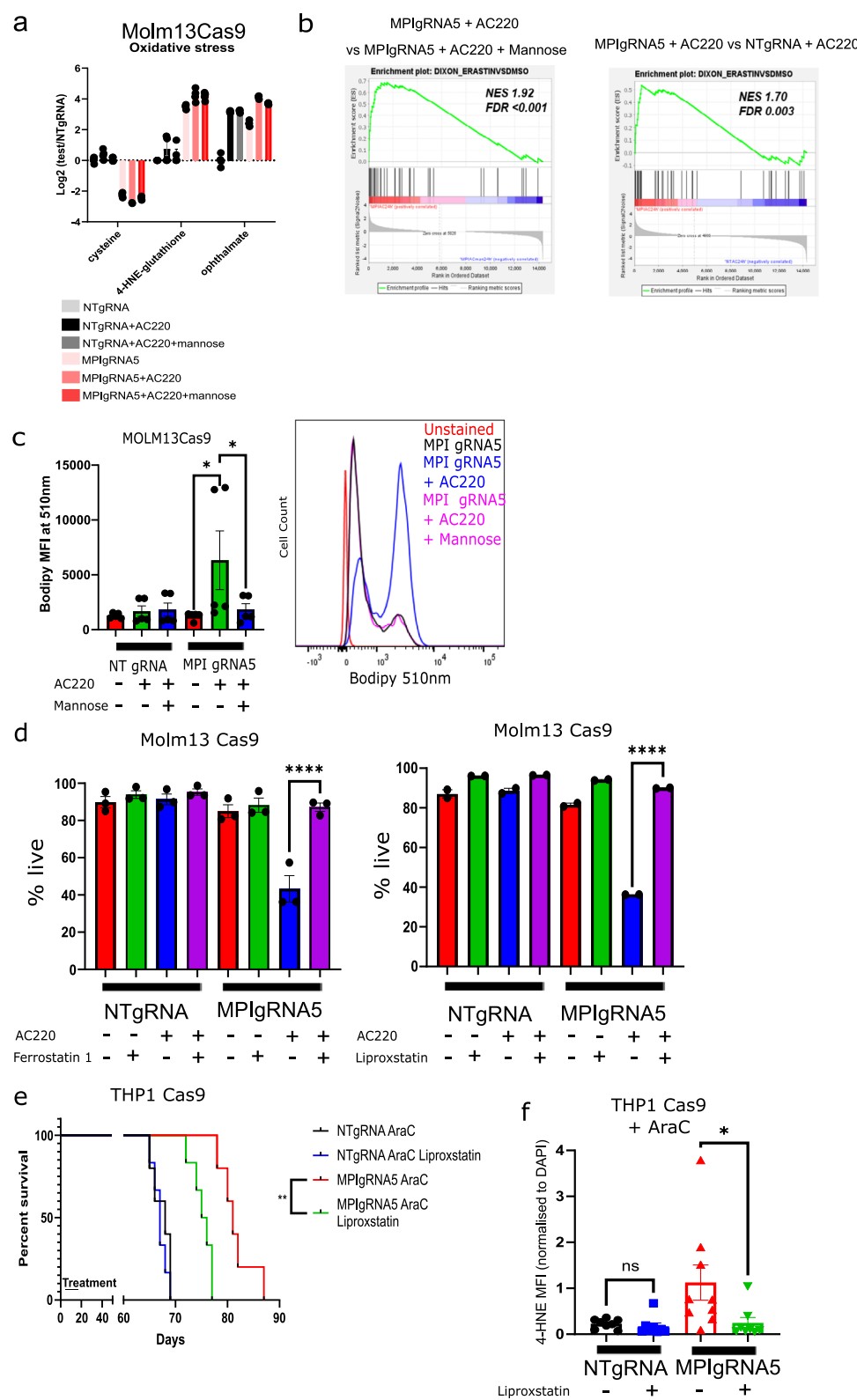

HSPC[18]. Elevated mannose metabolism may be more critical for developing tissues, as suggested by a recent report using hypomorphic alleles of PMM2, the enzymatic step following MPI in MM, that demonstrated that, while mannose sugar donors are necessary for embryonic development, they are only needed at very low levels after development[49]. Thus, the activity of MPI might be crucial for rapidly dividing cells but less essential in adult established tissues, a finding not uncommon in drug development where often embryonically lethal/essential genes are shown to be good drug targets with acceptable toxicity in the adult/homeostatic context[50].

MPI validity as a therapeutic target has been related to its ability to detoxify exogenous high levels of mannose[23,51]. However our mechanistic studies show that, even in the presence of physiological mannose concentrations, MPI inhibition can increase the toxicity of FLT3-TKI or AraC to AML cells due to defective fatty acid metabolism. While other reports on the role of MPI in cancer have established its

**Fig. 6 | MPI KO drives lipid peroxidation and ferroptotic cell death in AML cells. a** Levels of ROS and lipid peroxidation linked metabolites from global metabolomics profiling of NT gRNA and MPI gRNA5 treated with vehicle, AC220 (1 nM), mannose (100 μM) or in combinations as indicated for 48 h; (**b**) GSEA for an erastin induced (ferroptotic) gene signature[45] from RNA-seq comparing MPI gRNA5 with AC220 (1 nM) to MPI gRNA5 with AC220 (1 nM) and mannose (100 μM) (left) and MPI gRNA5 with AC220 to NT gRNA with AC220 (right). FDR and NES from 1000 permutations; (**c**) Mean fluorescence intensity at 510 nm of Bodipy 581/591 C11, which shows level of lipid peroxidation, in NT gRNA and MPI gRNA5 MOLM13 cells treated with AC220 (1 nM), mannose (100 μM) or in combinations as indicated 72 h after treatment. $N = 3$, 1 way Anova with Tukey's correction for multiple comparisons (left). Representative plots of flow cytometry fluorescence intensity plot of MPI gRNA5 with AC220 (1 nM), mannose (100 μM) or in combinations as indicated along with an unstained MPI gRNA5 control at 510 nm (right); (**d**) Percentage of live NT gRNA and MPI gRNA5 MOLM13 cells treated with vehicle, AC220 (1 nM),

mannose (100 μM), ferrostatin-1 (5 μM, left), liproxstatin (1 μM, right) or combinations as indicated for 6 days. $N = 3$ for ferrostatin, $N = 2$ for liproxstatin, 1 way Anova with Tukey's correction for multiple comparisons. **e** Kaplan-Meier survival curve showing survival time of mice after transplantation with THP1 NT gRNA or MPI gRNA5 cells, treated with cytarabine (50 mg/kg) for 7 days concurrent with or without liproxstatin treatment (10 mg/kg) for 10 days. Log-rank (Mantel-Cox) test; (**f**) Mean fluorescence intensity of 4-hydroxynonenal normalised to DAPI fluorescence from immunofluorescence images of THP-1 cells sorted from mice after transplantation with THP1 NT gRNA or MPI gRNA5 cells and soon after treatment with cytarabine (50 mg/kg) for 7 days concurrent with or without liproxstatin treatment (10 mg/kg) for 10 days. $N = 2$ fields of view from 4 animals in all samples except MPIgRNA5 with AraC where 5 animals were used. 1 way Anova with Tukey's correction. For all panels, ns = not significant, *=$p < 0.05$, **=$p < 0.01$, ***=$p < 0.005$, ****=$p < 0.001$. Source data are provided as a Source Data file. All data presented as mean ± SEM.

role in supporting or rewiring central carbon/glucose metabolism, none had previously unveiled a connection between MM and fatty acid metabolism, although the ability of exogenous mannose to upregulate FAO has been observed in T cells[52]. We show that the effects on fatty acid metabolism are prompted by reduction in protein N-glycosylation, a known consequence of MPI depletion, which likely drives the preferential activation of the ATF6 arm of the UPR[32]. Interestingly reduced protein N-Glycosylation has already been shown to be a therapeutic vulnerability in solid cancers[22] and leukaemia[53] and specifically in FLT3[ITD] AML due to reduced surface FLT3 expression upon glycosylation inhibition[54]. While we could show a pattern of reduced protein glycosylation in our models, we did not detect consistent findings for reduced FLT3 expression (Supplementary Fig. 10) thus making this mechanism less likely. On the other hand, we observed that activation of the ATF6 arm of the UPR led to downregulation of FAO in AML MPI KO cells through transcriptional downregulation of PPARA and other key FAO genes. FAO has emerged as a therapeutic target in AML[11,55] and is important in resistance to both standard[13] and novel therapies[56] via its ability to support oxidative phosphorylation. Therefore our findings support the role of FAO as a resistance mechanism in AML. Although FLT3-TKI and AraC have different mode of actions, resistance to both these drugs relies on enhanced FAO[13,57,58] and likely explains why MPI KO by causing defective FAO and increased PUFAs levels leads to sensitization to both therapies. Moreover the role of reduced glycosylation and ATF6 activation in driving these effects further suggest that the main role of MPI in our systems is the production of mannose from glucose, rather than the opposite, and explains why MPI inhibition was able to cause cell death in the absence of high mannose concentration.

Although the effects of MPI KO on cell death could be partly driven by defective TCA cycle activity leading to ATP depletion, we did not rescue the effects of MPI KO by supplementing the TCA cycle intermediate DMaKG. Instead AML MPI KO cells, particularly in the presence of FLT3-TKI, displayed features consistent with a state primed for ferroptotic cell death. These included reduced FAO, increased CD36 expression and PUFA uptake, reduced expression of SLC7A11, the cysteine transporter and target of the ferroptotic inducing agent erastin[59] and increased sensitivity to ferroptosis inducing agents and PUFA-induced toxicity. Ferroptosis is a non-apoptotic, iron dependent form of regulated cell death that is specifically characterised by the accumulation of lipid peroxides[60]. Susceptibility to ferroptosis has been shown in solid cancer models to be a feature of cells in a therapy resistant state because of their high dependency on the lipid hydroperoxides quenching proteins such as glutathione peroxidase 4[61]. Reduced FAO and downregulation of PPARA the master regulator of lipid catabolism have also been linked to susceptibility to ferroptosis in other cancer models[41,42,62] while increased uptake of PUFA via CD36 has been shown to trigger ferroptotic cell death/susceptibility in T cells[63,64]. In this respect, it is worth noting that increased

lipid uptake can enrich cancer cells membranes in PUFA since in contrast to de-novo synthesitized FA, diet and stroma derived FA are enriched for PUFA thus increasing susceptibility to ferroptosis[65]. Moreover, there is evidence that PUFA are most readily beta-oxidised in mammalian cells compared to SFA[66]. Therefore, a combination of increased lipid uptake and reduced FAO will lead to preferential accumulation of PUFA with an increased tendency towards lipid peroxidation, as seen in our MPI KO cells. Our rescue experiments with radical trapping agents confirm that ferroptosis is mostly driving cell death in MPI KO cells and future studies will be needed to further define the molecular mechanisms through which MPI KO drives increased PUFA uptake, CD36 expression and reduced cysteine uptake and SLC7A11 expression. Moreover, our findings also suggest that, similarly to persistent cells in solid cancers[61], triggering ferroptosis can be leveraged to drive cell death in therapy-resistant AML cells. However, the role of ferroptosis susceptibility in AML therapy resistant/ persister cells will require further studies.

In conclusion, our work supports the role of metabolic rewiring in driving therapy resistance in AML and demonstrates that targeting MPI and MM sensitizes AML cells to AraC and FLT3-TKI. Mechanistically we unveil a connection between MM and fatty acid metabolism, via preferential activation of the ATF6 arm of the UPR, leading to cellular PUFA accumulation, lipid peroxidation and ferroptotic cell death (Fig. 8). Finally, our findings also suggest that triggering ferroptosis could be used therapeutically to eradicate therapy-resistant AML cells.

## Methods

### Ethics statement
All work involving mice was performed under UK Home Office Personal Project License Number PP4153210 (Protocol 4) with further ethical approval for individual studies granted by our in vivo work establishment (BSU) Charterhouse Square Queen Mary University of London. In all experiments the maximum severity allowed under the project license was not exceeded. All work involving human samples was covered under ethics approval granted by East of England Cambridge Research Ethics Committee under the title "Barts Haemato-Oncology Research Tissue Bank" REC ref. 21/EE/0123, IRAS project ID 283103.

### Cell culture
MOLM13, MV411 and THP-1 cells were obtained from the Sanger Institute cancer cell collection and were grown in 1640-RPMI (Gibco, 21875034) supplemented with 10% FBS (Sigma, F4135) and 1% penicillin/streptomycin (Gibco, 15140122) at 37 °C with 5% $CO_2$, unless otherwise stated. For SEAHORSE analysis, media comprised of SEAHORSE RPMI (Agilent), without bicarbonate and FBS, supplemented with 11 mM glucose and 2mM L-Glutamine (SEAHORSE media), unless otherwise stated. HEK-293T Phoenix cells were obtained from the Huntly lab and grown in DMEM (Gibco, 41966029) supplemented with 10% FBS (Sigma, F4135) and 1% penicillin/streptomycin (Gibco,

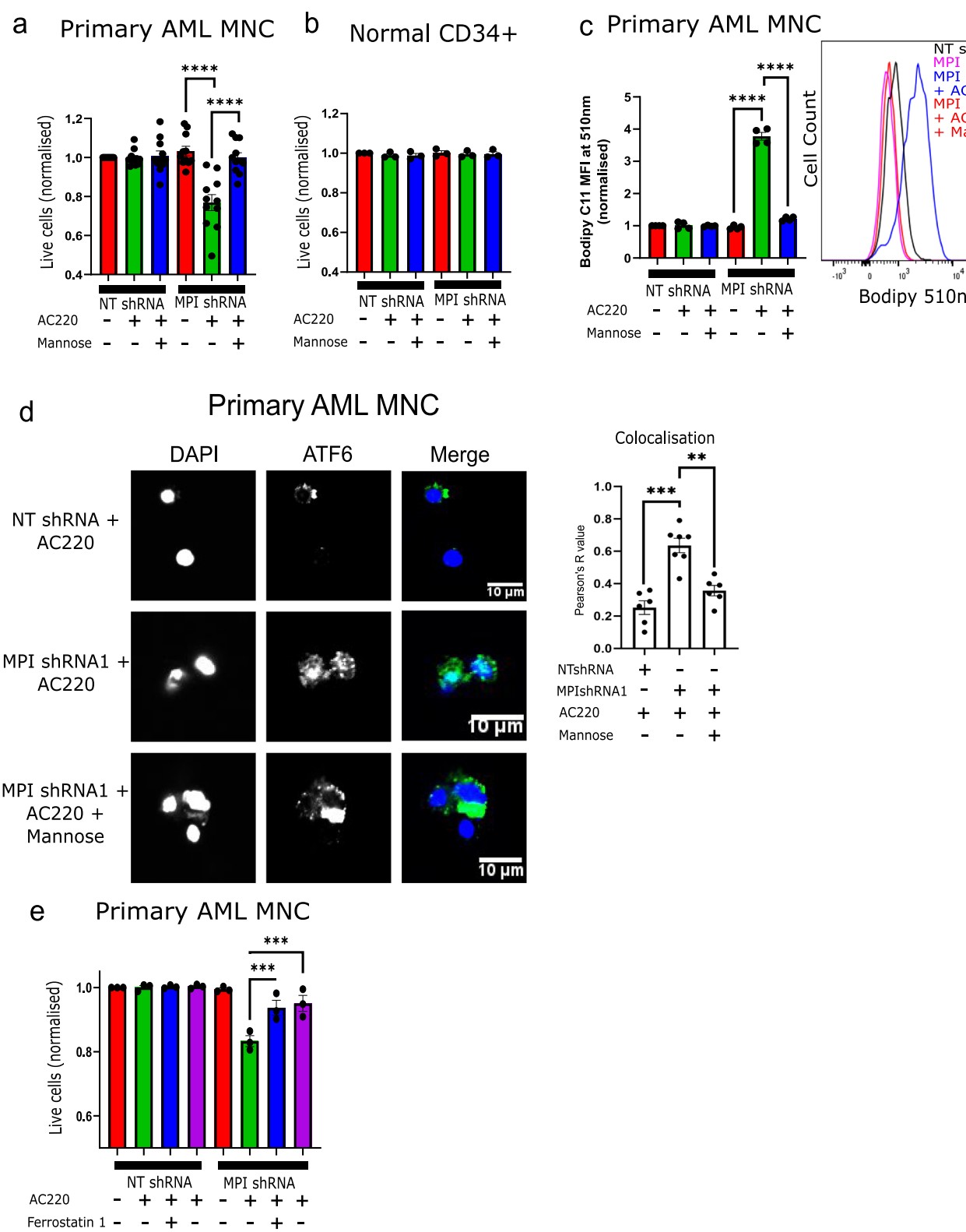

**a** Primary AML MNC

**b** Normal CD34+

**c** Primary AML MNC

**d** Primary AML MNC

DAPI    ATF6    Merge

NT shRNA + AC220

MPI shRNA1 + AC220

MPI shRNA1 + AC220 + Mannose

Colocalisation

**e** Primary AML MNC

15140122) at 37 °C with 5% $COO_2$ unless otherwise stated. For experiments where delipidated FBS was used, this was done by the adding of 2 g of fumed silica to 100 ml of heat inactivated FBS (Sigma, F4135) and stirring overnight at room temperature. The mixture was then centrifuged at 4000 × g for 25 min to pellet the silica and the supernatant retained. The supernatant was then filtered through a 0.2 μm filter to ensure sterility before storage at −20 °C for further use. Delipidated

FBS was used in RPMI at a final concentration of 10%. Cells were only kept in delipidated FBS for 3 days as any longer culture of cells in these conditions lead to large scale cell death in all conditions.

**Lentiviral transduction**

293T-Pheonix cells were grown in DMEM (Gibco) containing 10% FBS and transfected with construct plasmid (NTshRNA and MPI shRNA1

**Fig. 7 | MPI depletion in primary FLT3ITD AML samples causes ATF6 activation, lipid peroxidation and sensitization to FLT3-TKI therapy. a** Percentage of live cells of NT shRNA and MPI shRNA1 primary FLT3$^{ITD}$ AML MNC treated with vehicle, AC220 (2.5 nM) or AC220 (2.5 nM) and mannose (100 μM), as indicated normalised to NT shRNA with vehicle for each sample. Treated for 3 days, $N=11$, 1 way Anova with Tukey's correction for multiple comparisons; (**b**) Percentage of live cells of NT shRNA and MPI shRNA1 normal CD34 + primary cells treated with vehicle, AC220 (2.5 nM) or AC220 (2.5 nM) and mannose (100 μM), as indicated normalised to NT shRNA with vehicle for each sample. Treated for 3 days, $N=3$; (**c**) Mean fluorescence intensity at 510 nm of Bodipy 581/591 C11, which shows level of lipid peroxidation, in NT shRNA and MPI shRNA1 FLT3$^{ITD}$ primary AML MNC treated with vehicle, AC220 (2.5 nM) or AC220 (2.5 nM) and mannose (100 μM), as indicated normalised to NT shRNA with vehicle at 72 h after treatment. $N=4$, 1 way Anova with Tukey's correction for multiple comparisons (left). Representative plots of

flow cytometry fluorescence intensity plot NT shRNA and MPI shRNA1 with AC220 (2.5 nM) and mannose (100 μM) as indicated at 510 nm (right); (**d**) Confocal microscopy images of FLT3$^{ITD}$ primary AML MNC stained for DAPI (blue, 1st column) and ATF6 (green, 2nd column) with a merged image (3rd column), treated with vehicle, AC220 (2.5 nM) and mannose (100 μM) combinations as indicated 72 h after treatment with colocalisation analysis (right graph). $N=6$ for all except MPIshRNA with AC220 where $N=7$. 1 way Anova with Tukey's correction; (**e**) Percentage of live cells of NT shRNA and MPI shRNA1 primary FLT3$^{ITD}$ AML MNC treated with vehicle, AC220 (2.5 nM), Ferrostatin-1 (5 μM) or liproxstatin (1 μM) as indicated, normalised to NT shRNA with vehicle for each sample. Treated for 3 days, $N=3$, 1 way Anova with Tukey's correction for multiple comparisons. For all panels, ns = not significant, *=$p < 0.05$, **=$p < 0.01$, ***=$p < 0.005$, ****=$p < 0.001$. Source data are provided as a Source Data file. All data presented as mean values ± SEM.

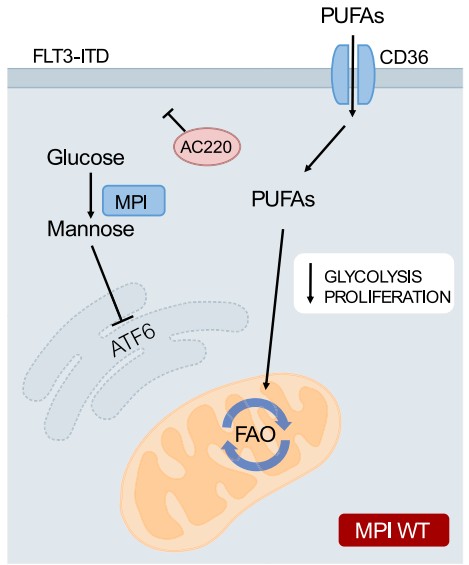
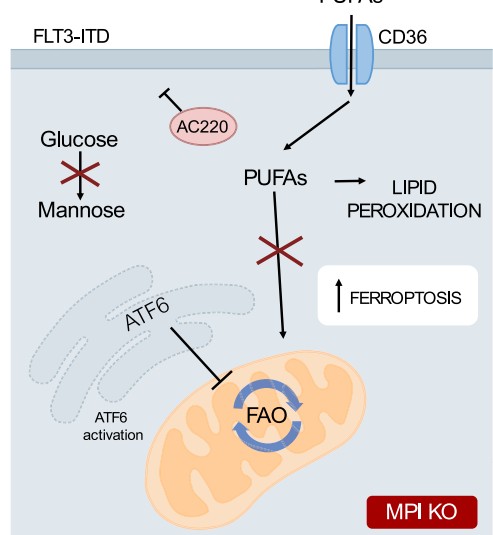

**Fig. 8 | Model: loss of MPI leads to cell death in AML through inhibition of FAO leading to PUFA accumulation and ferroptosis.** A model of the proposed mechanism. WT AML cells treated with both standard and FLT3 inhibitor therapies are able to escape cell death by adapting their metabolism, in this case by switching from glycolysis to fatty acid oxidation. Conversely treated AML cells with depleted MPI have preferential activation of the ATF6 arm of the unfolded proteins response

which inhibits fatty acid oxidation. This is paired with increased uptake of fatty acids, particularly polyunsaturated fatty acids, by MPI depleted cells. Both these effects lead to intracellular accumulation of PUFAs and PUFAs containing lipid species. These undergo lipid peroxidation leading to ferroptotic cell death in these cells.

and 2 TRIPZ inducible shRNA, Horizon Discovery, ATF6-YFP reporter fig. S5f), packaging construct (psPAX) and glycoprotein (pMD2G) using lipofectamine (Gibco, 18324010) for transient transfection. After 24 h, medium was replaced by RPMI with 5% FBS before overnight incubation at 32 °C with 5% $CO_2$. Media containing virus was harvested and filtered through a 0.45 μm filter to remove any cells 24 and 48 h post media change and either used immediately or stored at −80 °C in 1 ml aliquots for further use.

For transduction of cell lines, 1 million cells were seeded in 1 ml RMPI containing virus with 4 μg/ml polybrene (Santa Cruz Biotechnology, TR1003) and spun for 90 min at 900 × g on 2 consecutive days. Cells were then placed in an incubator at 32 °C for 4 h before the addition of a further 3 ml of RPMI with 5% FBS. Cells were then placed in an incubator at 37 °C for 24 h before the virus was removed by washing three times with sterile PBS.

Primary samples were cultured in StemSpan II (STEMCELL-Technologies, 09605) supplemented with 150 ng/ml SCF (Biolegend, 573908), 150 ng/ml Flt-3 (Biolegend, 550602), 10 ng/ml IL-6 (Biolegend, 570802), 25 ng/ml G-CSF (Biolegend, 578602), 20 ng/ml TPO (Biolegend, 763702), 1% HEPES (Gibco, 15630056) and 1% Penicillin/Streptomycin. All samples were cultured at 37 °C with 5% $CO_2$. Cells were not spun and instead viral supernatant was added to a final

volume of 10% of total culture volume. Samples were then cultured overnight before repeating. Virus was removed by washing three times with sterile PBS before continuing with culture as detailed below.

For MPI and NTshRNA transduced samples, cells were cultured in normal media with the addition of puromycin (1 μg/ml final concentration, ThermoFisher, A1113803) for selection of transduced cells and doxycycline (1 μg/ml final concentration, Sigma, D3072) for activation of the shRNA construct. Activation of the construct and transduction efficiency was determined by the percentage of RFP + cells. Cells were then cultured as normal with the addition of 1 μg/ml doxycycline for 72 h before plating for experiments to induce knockdown of the target genes.

For the ATF6 reporter, transduced cells were selected by treating cells with 1 μM tunicamycin (Tocris, 3516) to induce YFP expression in transduced cells before cell sorting for the YFP positive population. Cells were then cultured as normal.

## Human samples

The use of human samples, collection and publication of individual-level clinical data was approved by the East of England - Cambridge Research Ethics Committee REC 21/EE/0123. The conduct of the study was fully compliant with the ethical approval and sex and gender

information was not available for the samples used. Bone marrow and peripheral blood samples containing >80% blasts were collected from adult patients with AML at diagnosis or relapse and cryopreserved after mononuclear cell (MNCs) isolation using Lymphoprep™ (07851, STEMCELL Technologies) based density gradient centrifugation. Normal CD34⁺ enriched samples were taken from myeloma patients at the time of peripheral blood stem cell collections. Samples were thawed and transduced via lentivirus with NTshRNA or MPIshRNA as described above.

After transduction was confirmed by the presence of RFP expression by flow cytometry, primary samples were cultured on a stroma of MS-5 mouse cells that had been irradiated in H5100 media (STEMCELL Technologies) supplemented with penicillin/streptomycin, interleukin-3, G-CSF and TPO at 20 ng/ml final. For all assays, AML cells were removed from the co-culture and cultured without stroma. Assays were performed as described in viability section, however less cells were used in each case as numbers of cells in the primary samples was limited. To ensure that only transduced primary cells were removed and used for assays, RFP-positive percentage was assessed to be above 95%.

### Generation of CRISPR knockout clones
The knock-out of genes was accomplished using the CRISPR/Cas9 system. For this, MOLM13, MV411 and THP-1 cell lines were transduced with Cas9 lentivirus (Addgene, lentiCas9-Blast, #52962), generating Cas9 expressing cell lines. Functional gRNA sequences were obtained from genome-wide gRNA library and ligated in the backbone, conjugating MPI gRNA with mCherry (Addgene, pKLV2-U6gRNA5(Bbsl)-PGKpuro2AmCherry-W, #67977). Virus for Cas9, MPI gRNA and NT gRNA were produced and transduced as described above. Obtaining single cell colonies was performed using methylcellulose media (H4531; STEMCELL Technologies, Cambridge, UK), where cells were cultured in semi-liquid conditions to minimize movement. Seeding cells in methylcellulose in a low density (~500 cells/mL) permitted separate colonies to form from a single cell. After picking the colonies from the assay and culturing them in small volumes, the generation of cell lines from a single parent all with the same construct was realized.

### CRISPR KO gRNA sequences
*NT*gRNA - ATTTTCGTACCCTGGGACGC
*MPI*gRNA2 - AGGATCTTGGCATCCCCTC
*MPI*gRNA5 – CAATCAGGAACTGAAACTC

### SiRNA transfection
ATF6 (Ambion, AM16708) and non-silencing control (Ambion, 4390843) siRNAs were resuspended in nuclease free water and stored at −20 °C until further use. Molm13 cells were replated at a density of 200,000 cells/ml in normal RPMI with 10% FBS and 1% pen/strep 48 h before transfection. On day of transfection cells were pelleted, washed twice with PBS and finally resuspended in 100 µl buffer SF (Lonza, V4XC-2012). Cells were transferred to Lonza Nucelocuvettes and placed into the Lonza Nucelofector 4D X unit and transfected using program DJ-100. Cells were then removed and added to 2 ml of pre-warmed RPMI with 10% FBS and 1% pen/strep. Cells were taken for RNA extraction and qPCR analysis after 48 h and plated for annexin analysis at the same time.

### Treatments
AC220 (Insight Biotechnology, HY13001) was dissolved in DMSO and cells were treated at 1 nM final concentration, unless otherwise stated. Mannose (Sigma, 112585) was dissolved in sterile water and cells were treated at 100 µM final concentration, unless otherwise stated. Etomoxir (Cayman Chemicals, 11969) was dissolved in DMSO and cells were treated at 10 µM final concentration for all apoptosis and lipid peroxidation assays and 50 µM for all SEAHORSE experiments, unless otherwise stated. MLS0315771 (Insight Biotechnology, HY112945) was dissolved in DMSO and cells were treated at a final concentration of

1 µM, unless otherwise stated. Ceapin A7 (Sigma, SML2330) was dissolved in DMSO and cells were treated at a final concentration of 1 µM, unless otherwise stated. Fenofibrate (Sigma, F6020) was dissolved in DMSO and cells were treated at a final concentration at 10 µM, unless otherwise stated. Ferrostatin-1 (Sigma, SML0583) was dissolved in DMSO and cells were treated at a final concentration of 5 µM, unless otherwise stated. Erastin (Sigma, E7781) was dissolved in DMSO and cells were treated at a final concentration of 4 µM, unless otherwise stated. Sulfosuccinimidyl Oleate (SSO, Cayman Chemicals, 11211) was dissolved in DMSO and cells treated at a final concentration of 100 µM. Arachidonic acid (Sigma, 10931), palmitic acid (Acros organics, 129702500) and oleate (Sigma, O7501) were dissolved in isopropanol, with cells treated in concentrations as outlined below. AA147 (Tocris, 6759) was dissolved in DMSO and cells were treated at a final concentration of 5 µM. Tunicamycin (Tocris, 3516) was dissolved in DMSO and cells were treated at a final concentration of 1 µM. ZVAD (Sigma, V116) was dissolved in DMSO and cells were treated at a final concentration of 20 µM. Necrostatin-1 (Sigma, N9037) was dissolved in DMSO and cells were treated at a final concentration of 20 µM. GSK2656157 (SelleckChem, S7033) was dissolved in DMSO and cells were treated at a final concentration of 200 nM. MKC-3946 (SelleckChem, S8286) was dissolved in DMSO and cells were treated at a final concentration of 1 µM. Dimethyl alpha-ketoglutarate (Sigma, 349631) was used at a final concentration of 4 mM. RSL3 (SelleckChem, S8155) was dissolved in DMS and used at a final concentration of 100 nM unless otherwise stated.

### Viability assay
Around 1 million cells per condition were harvested by centrifugation at 500xg, before washing twice with PBS at 4 °C. Cells were then resuspended in annexin V binding buffer (BioWorld, 21720002-1) and annexin V-FITC (BioLegend, 640945, 1:150) and 7-AAD (BD Biosciences, 559925, 1:150) were added. Positive control cells for single colour compensation of the FITC and 7-AAD were generated by heating cell suspensions for 3 min at 70 °C. Cells were incubated for at least 15 min at 4 °C, protected from light. For analysis of live cells by DAPI exclusion cells were harvested as described above and resuspended in PBS. DAPI at a final concentration of 0.5 µg/ml was added. Cells were then kept at 4 °C protected from light until analysis. Cells were then analysed using LSR Fortessa (BD Biosciences) instruments. Downstream analysis was performed using FlowJo version 10.6.1 (BD Biosciences). Example FACS gating strategies are available in Supplementary Information.

### Cell counting
Cells were counted either using a CASY cell counter (Cambridge Biosciences) or a LUNA automated cell counter (Logos Biosystems). For LUNA cell counting, cell samples were mixed in a 1:1 ratio with 0.4% trypan blue solution.

### Competition assays
125000 Cas9/KO-mCherry (NTgRNA or MPIgRNA) per ml were added along with 125000 Cas9 cells with no mCherry (parental) per ml and were cultured in the presence or absence of AC220 (1 nM). At days 3, 6 and 9 the ratio between mCherry positive and negative cells were measured by flow cytometry using LSR Fortessa (BD Biosciences) instruments, with data normalised to the NT:parental ratio at day 3. Downstream analysis was performed using FlowJo version 10.6.1 (BD Biosciences). Example FACS gating strategies are available in the Supplementary Information.

### IC50 calculations
50000 NT gRNA, MPI gRNA2 and MPI gRNA5 MOLM13 cells were seeded and treated with increasing concentrations of AC220 in triplicate. After 3 days cell proliferation was measured using CellTiter 96

AQueous Non-Radioactive Cell Proliferation Assay (Promega, G5421) and normalised to cells treated with vehicle only as 100% proliferation.

## SEAHORSE MitoStress test

The SEAHORSE MitoStress test was performed essentially per manufacturer's instructions. In short, cells were grown and treated as described above. The SEAHORSE cartridge was hydrated with sterile water overnight in a 37 °C incubator with no $CO_2$, before the water being removed and replaced with SEAHORSE calibrant solution (Agilent, 100840-000) for at least 1 h before the cartridge is put into a SEAHORSE XF instrument. The cell plate was coated with Corning CellTak (Fisher, can no CB40240) adhesion solution at 22.4 μg/ml concentration in sodium bicarbonate at pH 8 for 15 min at room temperature before washing twice with sterile water. $10^5$ cells were harvested and washed with PBS before resuspension in 50 μl of prewarmed SEAHORSE RPMI media (Agilent, 103576-100) and adding onto the cell plate. Cells were adhered by centrifugation at $100 \times g$ for 1 min without braking, before incubation at 37 °C for 15 min in an incubator with no $CO_2$. 130 μl of additional SEAHORSE media along with etomoxir at a final concentration of 50 μM (only to etomoxir treatment wells) was added to wells and incubated for a further 15 min. The drugs were added to the cartridge in the following manner: port A – oligomycin, 1 μM final concentration, port B – FCCP, 0.3 μM final concentration, port C – rotenone/antimycin A, 0.5 μM final concentration (Agilent, 103015-100), all of which were prepared in SEAHORSE media. The assay was set up on the SEAHORSE XF machine with a 2 min mixing, 2 min wait and 2 min measuring for each well at each timepoint.

For palmitate SEAHORSE assays it was performed essentially as above, but cells were cultured overnight in RPMI media lacking FBS and glutamine, supplemented with 50 μM palmitate-BSA or BSA (Sigma, A8806) only. SEAHORSE media was without glutamine and palmitate (50 μM) was added to the + palmitate conditions. The rest of the assay was run as described above.

## Bodipy uptake assay

C1-Bodipy C12 500/510 (Thermo Fisher, D3823) was added to cells in normal culture conditions at a final concentration of 1 μM, concurrent with other treatments before being placed in an incubator at 37 °C and 5% $CO_2$ for 24 h. Cells were then pelleted and washed twice with PBS before resuspension in PBS and flow cytometry analysis using LSR Fortessa (BD Biosciences) instruments. Downstream analysis was performed using FlowJo version 10.6.1 (BD Biosciences). Example FACS gating strategies are available in Supplementary Information.

## Bodipy lipid peroxidation assay

One million cells of each condition were harvested and the positive control was treated with 100 μM tetrabutyl hydroperoxide for the duration of the Bodipy labelling. Bodipy 581/591 lipid peroxidation sensor (Thermos Fisher, D3861) was added to a final concentration of 4 μM and cells were incubated for 45 min at 37 °C with gentle shaking, protected from light. Cells were then removed, washed twice with PBS and analysed by flow cytometry using LSR Fortessa (BD Biosciences) instruments. Downstream analysis was performed using FlowJo version 10.6.1 (BD Biosciences). Example FACS gating strategies are available in Supplementary Information.

## Bodipy neutral lipid stain

Cells were grown as normal with the addition of fatty acids-BSA or BSA alone as vehicle at the following concentrations: palmitic acid: 50 μM, oleate: 50 μM and arachidonic acid 1 μM for 24 h. Around 1 million cells of each condition were then harvested and washed twice in PBS, before resuspension in PBS and incubating the samples with 2.5 μg/ml Bodipy 493/503 (ThermoFisher, D3922) for 20 min at 37 °C with agitation. Samples were then washed twice with PBS and resuspended in PBS and then kept on ice before analysis by flow cytometry using LSR Fortessa

(BD Biosciences) instruments. For data interpretation, cells of all conditions were grown in normal media with BSA only. These cells were harvested and stained as above and the mean fluorescence intensity (MFI) was measured and subtracted from lipid supplemented conditions as a baseline of intracellular lipid content. Downstream analysis was performed using FlowJo version 10.6.1 (BD Biosciences). Example FACS gating strategies are available in Supplementary Information.

## Metabolomic analysis

Cells were treated with AC220 (1 nM) and mannose (100 μM) for 48 h before 30 million cells were harvested and washed. Pellets were snap frozen and sent to Metabolon (Morrisville, NC) for LC-MS analysis. After extraction of metabolites, samples were split into four aliquots to be analysed in acidic positive, acidic negative and basic negative eluted from a UPLC C18 column (Waters) with the fourth aliquot separated with negative ionisation on a UPLC HILIC column (Waters). All analysis was performed on a Q-Exactive mass spectrometer (Thermo Scientific) coupled to an ACQUITY UPLC system (Waters) coupled to a HESI-II electrospray ionisation source. MS operated at 35000 mass resolution with scan range between 70 and 1000 m/z. All extraction and analysis of RAW data was performed in house by Metabolon. Raw counts were normalised by Bradford protein concentration and then each biochemical is rescaled to the median levels of each biochemical across all 6 conditions which is set as reference. Z scores are then calculated as the difference between the observed value in one condition minus the mean, all divided by the standard deviation, i.e. $x$-μ/σ. Targeted metabolomic analysis following U-$^{13}C_6$-glucose labelling was performed as described previously[9].

## Palmitate labelling

Cells were cultured in Plasmax media (Ximbio, 156371) supplemented with 5% dialysed FBS (Sigma). U-$^{13}C_{16}$-Palmitic acid (CK Isotopes, CLM409) was dissolved in isopropanol and added to cell culture concurrent with treatments at a final concentration of 50 μM for 24 h. 500,000 cells were harvested washed twice with ice cold PBS before extraction in solvent containing 50% methanol (Fisher, 10675112), 30% acetonitrile (Fisher, 10407440) and 20% $H_2O$ (All HPLC grade, Fisher, 10777404). Samples were then centrifuged and the supernatant retained. Samples were kept at −80 °C until analysis.

For liquid chromatography a ZIC-pHILIC guard column (SeQuant, 20 × 2.1 mm) and ZIC-pHILIC column (SeQuant, 150 × 2.1 mm, 5 μm, Merck KGaA) was used on a Thermo Scientific UltiMate 3000 HPLC system. The aqueous mobile-phase solvent consisted of a 0.1% ammonium hydroxide −20 mM ammonium carbonate solution and and the organic mobile phase was acetonitrile. A linear biphasic LC gradient was executed, staring from 80% organic phase to 80% aqueous phase. The gradient took 15 min for a total run time of 22 min. The flow rate was set to 200 μl/min with column temperature being maintained at 45 °C. The mass spectrometer used was a qExactive Orbitrap Mass Spectrometer (Thermo Fisher Scientific). The MS operated in polarity switching mode. Pre-analysis calibration was carried out for both ionization mode using a custom CALMIX and a low m/z range tune file was used. Full scan (MS1) data was acquired during polarity switch in profile mode. The resolution was 35,000 (at m/z range 75–1000). Automatic gain control was set to target of $1\times10^6$ with a max fill time of 250 ms. The parameters for spray voltages are as follows: +4.5 kV (capillary +50 V, tube: +70 kV, skimmer: +20 V) and −3.5 kV (capillary −50 V, tube: −70 kV, skimmer: −20 V). The s-lens RF level was set to 50 for the front optics. The probe temperature was set at 50° C and capillary temperature set at 375° C The sheath gas flow rate was 25 and auxiliary gas flow rate was 15. Sweep gas flow was set to rate of 1. The mass accuracy was sub 5 ppm and data was acquired using Thermo Xcalibur 4.3.73.11 software.

The peak areas of each metabolite and their respective iso-topologues were quantified using Thermo Tracefinder 4.1. Metabolites were identified by accurate mass of the singly charged ion and by retention times of authentic standards on the pHILIC column. The commercially standard compound mix (Merck: MSMLS-1EA) had been analysed previously on our LCMS system to determine accurate ion masses and retention times. Data were processed and corrected for natural abundance through Autoplotter. (Pietzke and Vazquez, 2020, Cancer Metab. doi: 10.1186/s40170-020-00220-x.).

## Immunocytochemistry

Approximately 1 million cells for each condition were harvested and washed twice with PBS before fixing with 4% paraformaldehyde in PBS for 10 min at room temperature. Cells were then washed before per-meabilisation with 0.3% triton x-100 in PBS for 15 min at room tem-perature, before a PBS wash and blocking with 0.2% fish skin gelatin for 1 h at room temperature with agitation. Cells were incubated with primary antibody diluted in blocking solution (ATF6, 1:200, Santa Cruz biotech, SC166659) for 1 h at room temperature with agitation before 5 washes with PBS. Samples were then incubated with secondary anti-body (AlexFluor 488, goat anti-rabbit, 1:1000, Thermo, A32731) pro-tected from light before 5 washes with PBS. Glass slides were coated with CellTak solution (22.4 µg/ml) at room temperature for 15 min before washing the slides twice with PBS. Cells were then pipetted onto the CellTak coated slides and left for 30 min at room temperature before the addition of DAPI Fluoromount (Southern Biotech, 0100-20) solution and covering with coverslips. Slides were then kept at 4 °C until imaging. All imaging was done on a Zeiss LSM 710 scanning confocal microscope at ×40 or ×63 magnification. Further image analysis and processing was performed using ImageJ, including use of the Coloc2 plugin for colocalisation analysis. For 4-HNE staining, the procedure was performed essentially as above, with anti-4-HNE anti-serum (Alpha Diagnostic International, HNE11-S) as the primary anti-body. Further analysis was performed with the FIJI software package for ImageJ, using the Coloc2 plugin for colocalisation. For enhancing contrast, saturated pixels were set to 0.3 or 0.5% and was performed on the whole image at once. For images where more than 0.3 or 0.5% of pixels were already saturated, this contrast enhancement had no visi-ble effect.

## Staining of surface proteins for flow cytometry

Approximately 1 million cells for each condition were harvested and washed twice with PBS before resuspension in PBS. FC block (Biole-gend, 422301, 1:200) was then added for 20 min at room temperature before the addition of primary antibody at a 1:100 dilution before incubation at 4 °C for 30 min. Cells were then washed 3 times with PBS before resuspension in PBS. Samples were then stored either on ice or at 4 °C protected from light until analysis by flow cytometry using LSR Fortessa (BD Biosciences) instruments.

The following antibodies were used: anti-human CD45-FITC (BioLegend, 304005), anti-mouse CD45-APC (BioLegend, 157605), anti-SLC7a11 (SantaCruz Biotech, SC98552). Downstream analysis was performed using FlowJo version 10.6.1 (BD Biosciences). Example FACS gating strategies are available in Supplementary Information.

## Lectin staining of surface glycoproteins

Approximately 1 million cells for each condition were harvested and washed twice with PBS before resuspension in PBS. Lectin from *Lyco-persicon esculentum* (tomato) FITC conjugate (Sigma, L0401) was added in a 1:100 dilution before incubation for 1 h at 4 °C, protected from light. Cells were then washed twice with PBS, resuspended in PBS before flow cytometry analysis using LSR Fortessa (BD Biosciences) instruments. Downstream analysis was performed using FlowJo ver-sion 10.6.1 (BD Biosciences). Example FACS gating strategies are available in Supplementary Information.

## Proteostat staining

Approximately 1 million cells per condition were collected and washed twice with PBS before fixation for 10 min at room temperature by resuspension in 4% paraformaldehyde. Cells were then washed twice with PBS and permeabilised with 0.3% triton X-100 in PBS for 15 min before a further wash and resuspension in assay buffer from the pro-teostat protein aggregation kit (Enzo life sciences, ENZ51023) with proteostat stain (1:5000). Cells were then kept on ice before analysis. Example FACS gating strategies are available in Supplementary Information.

## Western blot analysis

In total, $10^6$ Cells were lysed in Cell Signalling technololgies lysis buffer (Cell Signalling) supplemented with protease and phospha-tase inhibitors. DNA was sheared by passing the sample through a fine gauge needle 8 times before centrifugation for 10 min at 14,000 x $g$. The supernatants were retained and stored at −20 °C before further use. NuPAGE sample buffer (Invitrogen) was added to samples before separation on 4-12% polyacrylamide gels (Invitrogen, NP0321). Proteins were then transferred to PVDF membranes fol-lowed by blocking with 5% milk/Tris Buffered Saline with 0.1% tween (TBS-T), incubation with primary antibodies in 5% BSA/TBS-T over-night at 4 °C then corresponding secondary antibodies in 5% milk/TBS-T for 1 h at room temperature before incubation of the mem-branes with ECL (BioRad,. 1705061) and visualisation using an Amersham Imager (GE Healthcare). Further analysis was performed in ImageJ. Primary antibodies were used at the following concentra-tions: ATF6 (Abcam, AB122897), 1:500, total Histone 3 (Cell Signal-ling, 44995), 1:5000, MPI (Santa Cruz Biotech, SC393477), 1:1000, PARP (Cell Signalling, 56255), 1:500, EIF2α (Cell Signalling, 9722), 1:1000, phosphor-EIF2α (Ser51, Cell Signalling, 9721), 1:1000. Uncropped and unprocessed scans of the blots are available in Supplementary Information.

## Quantitative RT-PCR (qPCR)

RNA was extracted from ~1 million cells using the RNeasy mini kit (Qiagen, 74106) as per the manufacturer's instructions. RNA con-centration was then measured using a nanodrop and immediately reverse transcribed using the High-Capacity cDNA Reverse Transcrip-tion Kit (ThermoFisher,. 4368814) as per the manufacturers intstruc-tions. cDNA was then stored at −20 °C until use. For qPCR 20 ng of cDNA was added to 5ul of PowerUp SYBR Green MasterMix (Ther-moFisher, A25742) with 250 nM of forward and reverse primers for the gene of interest. Samples in a 384 well plate were then analysed by BioRad CFX384 Real-Time System (BioRad) with the following proto-col: 95 °C for 2 min, 95 °C for 5 s, 60 °C for 30 s when the fluorescence was measured before returning to step 2 and repeating 39 times, 95 °C for 5 s, 65 °C for 5 s, fluorescence measurement, 95 °C for 30 s and holding at 16 °C forever.

## qPCR primer sequences

*CPT1A* - Forward: CAAACTGGACCGGGAGGAAA; Reverse: TGTGCTGG ATGGTGTCTGTC.
*PPARA* - Forward: AGCTGTCACCACAGTAGCTT; Reverse: GGAAC TCTTCAGATAACGGGCT.
*ACOX1* - Forward: CGCCGAGAGATCGAGAACAT; Reverse: GCACT TTTCCTGACAGCCAC
*MPI* - Forward: CTGCCGGGAAAGGCATACG; Reverse: AGCAATCC ACTGGCTTAGGG
*ERO1B* - Forward: AGAGAACTGTTTCAAGCCTCG; Reverse: TCCAG ACACAAACCTTCTAGCC
*HERPUD1* - Forward: CGAGATTGGTTGGATTGGACC; Reverse: CA CCCAACGTGATGCAGGTA
*XBP1* spliced − Forward: TTGCTGAAGAGGAGGCGGAA; Reverse: CTGCACCTGCTGCGGACTCAG

*XBP1* total – Forward: TTCCGGAGCTGGGTATCTCA; Reverse: GAAAGGGAACCCCCGTATCC

*ATF6* – Forward: ATGAAGTTGTGTCAGAGAACC; Reverse: CTCTTTAGCAGAAAATCCTAG

## RNA sequencing

Cells were treated with AC220 (1 nM) and mannose (100 μM) for 24 or 72 h as indicated. RNA was extracted and purified with the RNAeasy kit (Qiagen, 74106) as per manufacturer's instructions. Concentration was measured by NanoDrop (ThermoFisher). Ribosomal depletion was performed using the QIAseq FastSelect™ RNA Removal Kit−24samples; HUMAN (Qiagen: THS-001Z-24). Library preparation was performed with the dUTP directional kit NEBNext® Ultra8482 II Directional RNA Library Prep Kit for Illumina®. Samples were analysed on a NovaSeq6000 S1 flowcell as a paired end 150 bp (PE150) run. We followed recommended guidelines in the analysis of RNA-seq data for quality control, read mapping, quantification of gene expression, assessment of reproducibility among biological replicates, and differential gene expression (Conesa et al., 2016, Genome Biology, doi: 10.1186/s13059-016-0881-8.). First, TrimGalore 0.4.5 was used with default parameters and paired-end mode for quality-trimming and filtering of read sequences. Then, STAR 2.6.1 (Dobin et al, 2013, Bioinformatics, doi: 10.1093/bioinformatics/bts635) was used to align the reads to the reference human genome assembly GRCh38 using the Ensembl release 95 annotation as reference transcriptome. Proportion of uniquely mapped reads was >91% in all samples. FeatureCounts 1.6.3 was run on paired-end reads to count fragments in annotated gene features, with parameters 'p -T 4 -t exon -g gene_id' (Liao et al., 2014). The R/Bioconductor package DESeq2 was then used to perform differential gene expression analyses between samples and conditions (Love et al., 2014, Genome Biology, https://doi.org/10.1186/s13059-014-0550-8). Raw data is available at https://www.ebi.ac.uk/arrayexpress/experiments/E-MTAB-11750.

## Gene-set enrichment analysis (GSEA) analysis

RNA-sequencing data was analysed for GSEA using the Broad Institute software (http://software.broadinstitute.org/gsea/index.jsp). For GSEA analysis normalised read counts for all conditions were used and genes were ranked using the signal-to-noise metric and FDR and NES were calculated using 1000 gene-set permutation.

## Public dataset analysis

Public datasets used are GSE6891, E-TABM 1029, GSE12417, GSE15434, GSE13159, GSE10358, GSE37642, GSE76009, GSE30377, GSE83533 and RNA-Seq from AML TCGA (data obtained from https://www.cbioportal.org/), BeatAML dataset (data obtained http://www.vizome.org/) and from manuscript 10.1056/NEJMoa1808777 (NEJM WUSM) were used to perform various analysis. For MPI expression, normalised expression was used and compared between different genotypes (FLT3^{ITD} vs FLT3^{WT}) or patient samples (diagnosis vs relapse). For analysis of MPI level based on AML karyotype pre normalised data from GEO GSE147515 was plotted based on AML karyotype and MPI level. For GSEA analysis, within relevant datasets, the expression profile patients expressing high (i.e. above median) levels of MPI was compared to that of patients expressing low (i.e. below median) MPI levels. Genes were ranked using the signal-to-noise metric and FDR and NES were calculated using 1000 phenotype permutation.

## In vivo experiments

In total, $3 \times 10^6$ luciferase expressing MOLM13 cells transduced with non-targeting CRISPR guide (NT-gRNA) or MPI KO CRISPR guide (MPI-gRNA) were transplanted into male 8-12 week old NSG (NOD.Cg-*Prkdc^{scid} Il2rg^{tm1Wjl}*/SzJ) mice (Jackson Laboratory) which were sublethally irradiated (2 Gy). Mice were treated by oral gavage with AC220 (5 mg/kg) or vehicle (22% hydroxypropyl-β-cyclodextrin/0.3% DMSO)

for 5 days. Survival time was measured from time of transplantation until time mice had to be culled due to overt clinical symptoms. IVIS bioluminescence imaging (PerkinElmer) was used to confirm and track disease dissemination. In brief, D-luciferin (Cat#122799, Perkin Elmer) was administered by intraperitoneal (IP) injection as per the manufacturer's recommendation, followed by IVIS bioluminescence imaging under anaesthesia (Isoflurane). Bioluminescence as a surrogate for tumour burden, was quantified using Living Image Software (version 4.7.2, PerkinElmer).

A 8 week old male NBSGW (NOD.Cg-*Kit^{W-41J} Tyr^+ Prkdc^{scid} Il2rg^{tm1Wjl}*/ThomJ) mice (Jackson Laboratory) had either 5000 THP-1 NTgRNA or THP-1 MPIgRNA5 cells infused by intravenous injection via the tail vein. 1 week after this AML engraftment was measured by flow cytometry based on percentage of human CD45 + cells in the peripheral blood. The next day mice were treated with 50 mg/mg cytarabine (Sigma, 251010) in PBS or PBS alone (vehicle) for 7 days via intraperitoneal injection. Survival time was measured from time of transplantation until time mice had to be culled due to overt clinical symptoms.

Bone marrow tissue was harvested by dissection of the leg bones of the mouse before the bones were flushed with PBS to obtain the cells. After collection 2 ml of red blood cell lysis buffer (ThermoFisher, 00430054) was added and incubated for 5 min at room temperature before inactivation by the addition of 10 volumes of PBS. Cells were then pelleted for further flow cytometry analysis as outlined previously above.

10 week old male NBSGW (NOD.Cg-*Kit^{W-41J} Tyr^+ Prkdc^{scid} Il2rg^{tm1Wjl}*/ThomJ) mice (Jackson Laboratory) had either 5000 THP-1 NTgRNA or THP-1 MPIgRNA5 cells infused by intravenous injection via the tail vein. 1 week after this AML engraftment was measured by flow cytometry based on percentage of human CD45 + cells in the peripheral blood. The next day mice were treated with 50 mg/mg cytarabine (Sigma, 251010) in PBS for 7 days with or without liproxstatin (10 mg/kg) for 10 days via intraperitoneal injection. Survival time was measured from time of transplantation until time mice had to be culled due to overt clinical symptoms. For 4-HNE assessment a cohort of 3 mice in each treatment group was culled soon after treatment and bone marrow harvested. Thereafter human CD45 + leukaemic cells were sorted for immunocytochemistry as described above.

All mice were housed in IVCs on a 12 h light/dark cycle at an ambient temperature of 20 °C and humidity of 27–35%.

## Statistical analysis

All data was visualised in Prism 8.4 (Graph Pad), with statistical tests outlined in figure legends for individual analyses. All data are shown as mean ± standard error of the mean, unless otherwise stated.

## Reporting summary

Further information on research design is available in the Nature Portfolio Reporting Summary linked to this article.

# Data availability

The RNA-sequencing data generated in this study have been deposited in the ArrayExpress database and are available at https://www.ebi.ac.uk/arrayexpress/experiments/E-MTAB-11750. Untargeted metabolomics analysis is available in Supplemental Data 1. All data are available in the article, Supplementary Information and source data. Source data are provided with this paper.

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

## Acknowledgements

We wish to thank the Barts Cancer Institute tissue bank for sample collection and processing. This research was supported by the BCI Flow cytometry facility (CRUK Core Award C16420/A18066). This work was supported by the Wellcome Trust (PG, 109967/Z/15/Z), the American Society of Haematology (PG, Global Research Award) and Cancer Research UK (PG, Advanced Clinician Scientist fellowship, C57799/A27964). K.R-P. was supported by the Academy of Medical Sciences (SBF004\1099) J.H.M.P. was supported by a research grant from Science Foundation Ireland (SFI) under Grant Number 16/RC/3948 and co-funded under the European Regional Development Fund and by FutureNeuro industry partners. K.T. was funded by Wellcome Trust (Grant References: RG94424, RG83195, G106133), UKRI Medical Research Council (RG83195) and Leukaemia UK (G108148).

## Author contributions

P.G. and K.W. designed the experiments. K.W. and L.S.D. performed laboratory assays. K.M.R. G.V.H. designed and performed the palmitate labelling assay. A.M.S.M., C.P., V.D. and S.C.J. helped with experiments. L.N.L. and P.G. performed computational analysis of published datasets. P.M. performed RNA sequencing analysis. K.W., G.G., C.M., K.R.K. and R.A. designed, performed or helped in vivo studies. J.H.M.P. provided reagents and input for UPR analysis. C.P. and K.R-P. provided input on methodology, reagents, experimental design and analysis of UPR. K.T. and G.S.V. designed and performed original CRISPR screen. K.W. and P.G. wrote and edited the manuscript. P.G. designed the study and supervised the project with B.J.P.H. mentorship/support at the start. All authors critically reviewed the manuscript and approved the final version.

## Competing interests

The authors declare no competing interests.

## Additional information

**Peer review information** : *Nature Communications* thanks Eric Chevet, Courtney Jones and Silvia von Karstedt for their contribution to the peer review of this work. Peer reviewer reports are available.

