## [Peer Review File · Nature Communications]

Mannose metabolism inhibition sensitizes acute myeloid leukemia cells to therapy by driving ferroptotic cell deathREVIEWER COMMENTS

Reviewer #1 (Remarks to the Author):

Woodley et al. demonstrate that mannose metabolism protein MPI is elevated in AML, especially FLT3-ITD mutant AML, and correlates with a poor overall survival for AML patients. Knockout of MPI or treatment with the MPI inhibitor MLS0315771 sensitizes AML cell lines to FLT3 inhibitor AC220 and cytarabine in vitro and in vivo through the induction of ferroptotic cell death. Unbiased metabolomics analysis, gene expression analysis, seahorse, and ¹³C tracing experiments revealed that MPI knockout resulted in a decreased FAO. MPI knockout cells also have increased activation of ATF6 which decreases OXPHOS in AML cells through regulation of PPAR α . Finally, examination of primary AML MBC and normal CD34+ cells revealed that MPI shRNA knockdown decreased cell viability specifically in the AML cells, increased levels fatty acid peroxidation, and led to higher levels of nuclear ATF6. This reviewer is enthusiastic about this well written and novel study. There are a few experiments that appear to only have been performed in the Molm13 cell line. It would be very helpful to repeat these experiments in additional AML lines to increase the rigor of the overall study. Additional specific comments are listed below.

Major:

1. The ferrostatin and liproxstatin results are very striking. It appears that these experiments have only been done in the Molm13 line. Given the importance of this experiment for the overall conclusion that MPI is inducing ferroptotic cell death it would be helpful to repeat this experiment in the other AML lines and a small number of primary specimens.
2. Along a similar line, the rescue experiment with the PPAR α agonist helps to make the functional connection between ATF6 and FAO. It would be helpful to repeat this experiment in additional cell lines.
3. Figures 1 and 2 show that MPI expression correlates with a worse overall survival and is increased at relapse. Further, MPI knockdown sensitizes cells to chemotherapy. The mechanistic experiments are all performed with AC220. Does MPI knockdown sensitize AML cells to cytarabine through the same mechanism as the authors have shown for AC220?
4. Does MPI expression correlate with survival specifically in patients with a FLT3-ITD mutation? Perhaps it is an even better predictor of survival in this patient subset?
5. Loss of fatty acid oxidation has been described to target AML cells in various studies. Most of these reports have focused on the role of FAO in energy metabolism. Does adding back a cell permeable TCA cycle intermediate like dimethyl-ketoglutarate rescue or partially rescue the effect of MPI knockout?

Minor:

1. Using more distinct colors (or different shapes) would be helpful to see the differences in the PCA plots in supplemental figure 3B.

Reviewer #2 (Remarks to the Author):

NatComms_370192

In this manuscript, Woodley and colleagues establish a link between alteration of the mannose biosynthesis pathway and AML resistance to FLT3 inhibitors and cytarabine that requires the activation of the ATF6 arm of the unfolded protein response and rewiring of fatty acid metabolism. Although the manuscript presents interesting data, there are several points that need to be further documented or clarified in particular regarding UPR-related observations.

- 1) Indeed, the UPR is by essence a non-selective response in the present case activated by the accumulation of improperly folded proteins in the endoplasmic reticulum (ER) due to altered N-linked glycosylation. The authors state that "the ATF6 arm of the UPR (which) is particularly sensitive to changes in glycosylation", however, they refer to a manuscript in which the glycosylation levels of ATF6 itself is altered, which is obviously not the case in the data presented in the current manuscript as shown in Fig S5D. Indeed, ATF6 N-glycosylation levels are easily

detected using SDS-PAGE and western blotting. I am also a bit puzzled by ATF6 apparent molecular weight observed in fig S5D which does not correspond to most of the literature findings (for instance see PMID: 10564271). N-glycosylation of ATF6 could be controlled by the authors by using the N-glycosylation inhibitor tunicamycin.

2) If ATF6 is activated, the authors should show the cleavage of the proteins using western blot (of course positive controls should be included). How does fatty acid alteration impact on the functionality of the secretory pathway? At last, at present stage the immunofluorescence data are not convincing and should be further documented with higher magnification.

3) If there is a N-glycosylation defect as witnessed using lectin cell surface staining, the authors should demonstrate that this also correlates with an accumulation of improperly folded proteins in the ER (and the subsequent activation of the UPR).

4) Regarding the selectivity of the UPR, the authors should demonstrate that the other arms of the UPR are i) not activated and ii) not involved in the phenotype resistance observed in AML. For instance, PERK (EIF2AK3) and eIF2alpha phosphorylation status should be evaluated, as well as the activation of IRE1 and the non-conventional splicing of XBP1 mRNA as the latter has been demonstrated to contribute to lipid homeostasis.

5) It has been shown that the expression of XBP1u mRNA is under the control of the ATF6 arm (PMID:12586069), how do the authors reconcile this information with their model?

6) The use of pharmacological agents to modulate the ATF6 arm of the UPR should be i) confirmed in the cell system to prove that they either prevent or activate the cleavage of ATF6, ii) confirmed with other pharmacological agents (at least for the inhibition part) with for instance inhibition of ATF6 cleavage using serine protease inhibitors, and iii) also confirmed using genetic approaches (eg siRNA or overexpression of ATF6f).

7) Beyond ATF6, the other arms of the UPR were also shown to be linked to the control of ferroptosis activation and therefore should be documented in the authors' models.

Reviewer #3 (Remarks to the Author):

Woodley et al. identify that mannose-6-phosphate isomerase (MPI) knockout sensitized therapy resistant acute myeloid leukemia (AML) cells to FLT3-tyrosine kinase inhibitor (TKI) AC220 (quizartinib) and AraC. They attempt to identify by which mechanism MPI knockout sensitizes to conventional therapy and propose that increased fatty acid uptake (via increased CD36 expression) and unfolded protein response (UPR) leads to activation of ATF6 and subsequent reduced mitochondrial fatty acid oxidation resulting in increased amounts of free PUFAs promoting ferroptotic cell death under treatment with AC220 and AraC.

While the authors identify an interesting connection of MPI in the mannose pathway and therapy resistance in AML, the extent to which this relates to ferroptosis is entirely unclear. The mechanism of how MPI knockout increases fatty acid uptake and reduces FAO is not sufficiently connected to the mechanism of ferroptosis and misses important control *in vivo/in vitro* experiments.

Major points:

In Figure 2F: it is unclear whether the phenotype has anything to do with ferroptosis. Given the direction of the paper, they need to rescue AraC/AC220 treatment with Liproxstatin-1 to show that the effect is mediated by ferroptotic cell death and include 4-HNE stains *in vivo*.

What type of cell death do therapy (AC220, AraC)-sensitive parental AML cells undergo upon treatment, is it ferroptosis or other types of cell death? The mechanism of AC220 and AraC mediated therapy is very different, why would they have the same mechanism of resistance? They need to perform rescue experiments with Fer-1, zVAD, Nec-1 to rescue different types of cell death

Figure 6 is the only figure attempting to link the mechanism to ferroptosis. However, given that this is a prominent novelty of the manuscript, they need to include these validations in all mechanistic corner points of the manuscript, e.g. is the MPI ko lipid ROS phenotype rescued by ATF6/CD36/etc. inhibition? Can they rescue lipid ROS induction in MPI ko with Fer-1? What about the response of MPI ko cells to all commonly used ferroptosis inducers (RSL2, ML210. Etc.).

They see a difference in total lipid uptake, what about actual lipid peroxidation in MPI ko cells?

Which of the many things regulated in MPI ko cells is responsible for ferroptosis sensitization? Cd36 upregulation, xCT regulation, cystine uptake, differences in GSH levels?

Minor points:

Figure 1:

- A: labels need to be bigger
- F: not consistent with color scheme (MPI high red)

Figure 3:

- A: Why does mannose supplementation increase the amount of PUFA in WT cells?
- C: please show different FACS plot (half offset) for better overview. Should there not be an increase with AC220 which is then rescued with mannose?
- A and D do not show the same results, what is the normalization in A?

Ext. Figure 6:

- A/B: How do you explain that ac220 does not sensitize more to ferroptosis but rescue from it, like mannose and CD36 inhibition?

Figure 7:

- C: misleading description (not clear whether bodipy c11)

Dear Reviewers,

Please find below a point by point detailed rebuttal to your comments. Please note also that Extended Data Figure 6 has been split into 2 figures (Extended Data Figures 6 and 7) and Extended Data Figure 7 from the original submission has become Extended Data Figure 8 to accommodate additional experimental data. All changes have been highlighted in the manuscript and below here as required.

REVIEWER COMMENTS

Reviewer #1 (Remarks to the Author):

Woodley et al. demonstrate that mannose metabolism protein MPI is elevated in AML, especially FLT3-ITD mutant AML, and correlates with a poor overall survival for AML patients. Knockout of MPI or treatment with the MPI inhibitor MLS0315771 sensitizes AML cell lines to FLT3 inhibitor AC220 and cytarabine in vitro and in vivo through the induction of ferroptotic cell death. Unbiased metabolomics analysis, gene expression analysis, seahorse, and ¹³C tracing experiments revealed that MPI knockout resulted in a decreased FAO. MPI knockout cells also have increased activation of ATF6 which decreases OXPHOS in AML cells through regulation of PPAR α . Finally, examination of primary AML MBC and normal CD34+ cells revealed that MPI shRNA knockdown decreased cell viability specifically in the AML cells, increased levels fatty acid peroxidation, and led to higher levels of nuclear ATF6. This Reviewer is enthusiastic about this well written and novel study. There are a few experiments that appear to only have been performed in the Molm13 cell line. It would be very helpful to repeat these experiments in additional AML lines to increase the rigor of the overall study. Additional specific comments are listed below.

We thank the Reviewer for sharing their enthusiasm about our work. We agree that to increase the rigor of the overall study and its validity, we had to repeat some experiments in additional AML lines as also suggested by other Reviewers. We have now performed these experiments and discuss the main findings in response to the points below and highlight them in the manuscript.

Major:

1. The ferrostatin and liproxstatin results are very striking. It appears that these experiments have only been done in the Molm13 line. Given the importance of this experiment for the overall conclusion that MPI is inducing ferroptotic cell death it would be helpful to repeat this experiment in the other AML lines and a small number of primary specimens.

We have now repeated the ferrostatin and liproxstatin rescue experiments in both THP1 and MV411 cell lines. The effects on viability of MPI knock-out (KO) cells treated with cytarabine (AraC), in the FLT3 wild-type THP1 cells, or treated with AC220, in the FLT3-ITD mutant MV411 cells, were rescued by both ferrostatin and liproxstatin thus confirming that ferroptosis is driving cell death in MPI KO cells treated with both cytarabine and FLT3-inhibitors. Similar results were also observed in further 3 primary FLT3-ITD mutant AML MNC samples treated with AC220. These panels have now been added to Extended Data Fig. 6D and Figure 7E and the manuscript changed accordingly.

2. Along a similar line, the rescue experiment with the PPAR α agonist helps to make the functional connection between ATF6 and FAO. It would be helpful to repeat this experiment in additional cell lines.

We have now repeated these experiments in both MV411 and THP1 MPI KO cells treated with AC220 and AraC respectively and observe a similar rescue in viability upon treatment with fenofibrate. These panels have now been added to Extended Data Fig.4H

3. Figures 1 and 2 show that MPI expression correlates with a worse overall survival and is increased at relapse. Further, MPI knockdown sensitizes cells to chemotherapy. The mechanistic experiments are all performed with AC220. Does MPI knockdown sensitize AML cells to cytarabine through the same mechanism as the authors have shown for AC220?

We agree with the Reviewer that we should have made it clear that MPI KO sensitize cells to AraC and AC220 through a similar mechanism. Based on the additional experiments performed we now provide further evidence supporting this. We show that THP1 MPI KO cells, similarly to MPI KO FLT3-ITD mutant cells, have defective FAO, increased lipid peroxidation and undergo ferroptotic cell death (Extended Data Fig.4G-H, Figure 6E-F and Extended Data Fig.6D). It is also worth noting that there is significant literature supporting that resistance to AraC arises from increased reliance on oxidative phosphorylation (OxPhos) and FAO (Farge et al., *Cancer Discovery* 2017; Bosc et al., *Nature Cancer* 2021). Interestingly in our Seahorse assays (Extended Data Fig. 4F) THP1 control cells treated with AraC display also higher maximal oxygen consumption. As MPI KO impairs both FAO and OxPhos based on our data, it is therefore not surprising it can overcome the resistant phenotype of AraC treated cells. Moreover MPI KO leads to the accumulation of intracellular PUFAs through both defective FAO and increased uptake, thus sensitizing AML cells to ferroptosis. As a similar point has also been raised by Reviewer 3, we have added a sentence in the discussion regarding this.

4. Does MPI expression correlate with survival specifically in patients with a FLT3-ITD mutation? Perhaps it is an even better predictor of survival in this patient subset?

We thank the Reviewer for raising this interesting point. We have analysed the TCGA dataset by specifically looking at the FLT3-ITD mutant patients. Interestingly, although we see a clear difference in median survival with high expressing MPI patients having a shorter survival, this does not reach statistical significance possibly because of the limited numbers of patients in each group. We have added this panel in Extended Data Fig. 1F and added a comment regarding this in the manuscript. It is also worth noting however that the expression levels of MPI are higher in FLT3-ITD mutant patients (see Fig.1C and Extended Data Fig.1E) hence probably reducing the effects of low MPI expression levels when dichotomising patients based on the median MPI expression within the FLT3-ITD patient group. Interestingly, if we dichotomise FLT3-ITD patients based on median of expression in the overall TCGA cohort, we observe similar data but still not significant due to small numbers (shown both here for the Reviewers). Whether the effects of MPI expression levels on survival could be more obvious in patients treated with FLT3 inhibitors as our mechanistic

work might suggest, is a question we cannot answer as there are no publicly available RNA expression analysis in patients at diagnosis treated with FLT3 inhibitors.

FLT3-ITD patients only

5. Loss of fatty acid oxidation has been described to target AML cells in various studies. Most of these reports have focused on the role of FAO in energy metabolism. Does adding back a cell permeable TCA cycle intermediate like dimethyl-ketoglutarate rescue or partially rescue the effect of MPI knockout?

We thank the Reviewer for suggesting this very interesting experiment which we have now performed in 2 MPI KO cell lines, i.e. THP1 and MOLM13 (shown in Extended Data Fig. 4I). In both cell lines we have not seen a clear rescue in viability by dimethyl- α -ketoglutarate (DMaKG) thus suggesting that the defective FAO in MPI KO cells leads to cell death through effects not limited to the inhibition of OxPhos. Indeed our data suggest that by blocking FAO, MPI KO cells accumulate PUFAs leading to ferroptotic cell death. We speculate that DMaKG only rescues TCA cycle function and OxPhos but not the defective FAO and PUFAs accumulation (also due to increased uptake) in MPI KO cells and is therefore not able to provide a full rescue in viability for treated MPI KO cells. We have added a similar comment in the manuscript.

Minor:

1. Using more distinct colors (or different shapes) would be helpful to see the differences in the PCA plots in supplemental figure 3B.

We have now changed the colors in Extended Data Fig. 3B as requested.

Reviewer #2 (Remarks to the Author):

NatComms_370192

In this manuscript, Woodley and colleagues establish a link between alteration of the mannose biosynthesis pathway and AML resistance to FLT3 inhibitors and cytarabine that requires the activation of the ATF6 arm of the unfolded protein response and rewiring of fatty acid metabolism. Although the manuscript presents interesting data, there are several points that need to be further documented or clarified in particular regarding UPR-related observations.

1) Indeed, the UPR is by essence a non-selective response in the present case activated by the

accumulation of improperly folded proteins in the endoplasmic reticulum (ER) due to altered N-linked glycosylation. The authors state that “the ATF6 arm of the UPR (which) is particularly sensitive to changes in glycosylation”, however, they refer to a manuscript in which the glycosylation levels of ATF6 itself is altered, which is obviously not the case in the data presented in the current manuscript as shown in Fig S5D. Indeed, ATF6 N-glycosylation levels are easily detected using SDS-PAGE and western blotting. I am also a bit puzzled by ATF6 apparent molecular weight observed in fig S5D which does not correspond to most of the literature findings (for instance see PMID: 10564271). N-glycosylation of ATF6 could be controlled by the authors by using the N-glycosylation inhibitor tunicamycin.

We thank the Reviewer for raising this important point and agree that the UPR-related changes in our MPI KO cells required further experimental work. We have addressed this by performing Western Blot (WB) in our experimental conditions and in the presence of tunicamycin a known N-glycosylation inhibitor using different commercial ATF6 antibodies as those in the references provided are not commercially available (please note that the positive control used in Extended Data Fig.5E and G, left panel, was from OCI-AML3 lysates, while we have stronger positive control using MOLM13 in Extended Fig.5G right). We have found that, amongst many antibodies used, the Abcam anti-ATF6 antibody [1-7] (ab122897) is able to detect the cleaved (active) ATF6 form although not an obvious shift due to changes in N-glycosylation even with tunicamycin, possibly because the unglycosylated form might be rapidly cleaved. We now show in the new Extended Data Fig.5E that in MPI KO cells treated cells there is an increase in the ratio between cleaved/full length ATF6 rescued by mannose (with quantitation of all independent experiments). Overall we believe that the changes are consistent with MPI KO cells activating ATF6 through its cleavage and this being rescued by the addition of mannose. This corroborates similar evidence already provided through immunofluorescence and ATF6 reporter in Figure 5B-D and Extended Data Figure 5D. Interestingly in contrast to immunofluorescence but in keeping with the ATF6 reporter assay, the western blot show that AC220 treatment reduces ATF6 activation in both Control and MPI KO cells and this might reflect sensitivity of the assay and/or be related to the effects of AC220 on protein translation/ER protein load (due to its antiproliferative effects) and ATF6 activation at the assessed timepoint. We have made a comment regarding this in our manuscript.

2) If ATF6 is activated, the authors should show the cleavage of the proteins using western blot (of course positive controls should be included). How does fatty acid alteration impact on the functionality of the secretory pathway? At last, at present stage the immunofluorescence data are not convincing and should be further documented with higher magnification.

We refer to point 1 above with regards to showing the cleaved ATF6 form in our blots. We have also repeated the immunofluorescence and provided more convincing images in Fig 5C and Extended Data Fig.5D. With regards on how fatty acid alteration impact the secretory pathway, it is worth noting that changes in phospholipid production, particularly phosphatidylcholine, have been linked as a consequence of the specific activation of the ATF6 arm of the UPR leading to ER expansion in response to cargo load (see PMID:21521793 and PMID:19420237). We also observe a preferential activation of ATF6 and concomitant relative increase in phosphatidylcholine (Extended Data Fig.3E) in MPI KO cells. We did not measure if this was due to increased biosynthesis as was beyond the scope of our work, however it is indeed possible that changes in fatty acid metabolism due to

activation of ATF6 might have effects on ER expansion/biogenesis and secretory pathway. Indeed we see changes in expression levels of ATF6 target genes involved in the secretory pathway such as *ERO1B* and *HERPUD1* in MPI KO cells and following modulation with ATF6 inhibitor/activator (Extended Data Fig.5M).

3) If there is a N-glycosylation defect as witnessed using lectin cell surface staining, the authors should demonstrate that this also correlates with an accumulation of improperly folded proteins in the ER (and the subsequent activation of the UPR).

We have addressed this point using flow-cytometry staining with the ProteoStat aggresome detection kit (<https://www.enzolifesciences.com/ENZ-51035/proteostat-aggresome-detection-kit/>) which is able to detect accumulation of aggregated, misfolded protein in response to cellular stress. As shown in Extended Data Fig.5B, we can see an increased ProteoStat signal in MPI KO cells which is rescued by the addition of mannose. Interestingly the increase in misfolded protein is significant but marginal (less than 2 fold) which might explain the preferential activation of the ATF6 arm of the UPR (see response to point 5 below for more detailed explanation of this point).

4) Regarding the selectivity of the UPR, the authors should demonstrate that the other arms of the UPR are i) not activated and ii) not involved in the phenotype resistance observed in AML. For instance, PERK (EIF2AK3) and eIF2alpha phosphorylation status should be evaluated, as well as the activation of IRE1 and the non-conventional splicing of XBP1 mRNA as the latter has been demonstrated to contribute to lipid homeostasis.

We thank the Reviewer for this suggestion and agree that these experiments are needed to further clarify which arms of the UPR are driving the observed phenotype. We have now performed WB for total and phospho-eIF2alpha and qPCR for the total and spliced form of XBP1 (see schematic below for specific design of these primer sets, sequences are available in the manuscript) as read-out specifically for the PERK and IRE1 arms of the UPR. Interestingly we do not clearly observe a consistent upregulation of these branches of the UPR in our MPI KO cells when quantifying all the replicates (Extended Data Fig.5F). To further assess the functional role of these other branches of the UPR in the phenotypes observed, as also suggested by Reviewer 3, we have performed viability assays in MPI KO cells treated with AraC or AC220 after addition of either IRE1 or PERK inhibitor, MKC-3946 and GSK2656157 (Extended Data Fig.5L). In these experiments we did not observe any significant changes following the addition of PERK or IRE1 inhibitors thus suggesting that these branches of the UPR response do not play a functional role in the phenotype observed in MPI KO cells. This might also explain while adding more broad UPR activators partly phenocopies the effects of MPI even if likely activates all 3 branches as only ATF6 appears to play a functional role. We have now added a comment regarding these observations in the results section.

5) It has been shown that the expression of XBP1u mRNA is under the control of the ATF6 arm (PMID:12586069), how do the authors reconcile this information with their model?

We thank the Reviewer for pointing us towards these key papers characterising the UPR in mammalian cells (both PMID: 12586069 and PMID: 11779464 from the same authors and referred to in the suggested paper). They indeed show that ATF6 activation precedes the production of spliced XBP1 and indirectly suggests that activated ATF6 by binding to the ERSE (ER stress response element) controls the expression of XBP1u mRNA. However in both papers, IRE1 activation is still essential for the production of XBP1s. Interestingly we do observe that total XBP1 mRNA level, but not the spliced XBP1 mRNA level, is higher (hence the reduced ratio) in MPI KO cells (Extended Data Fig.5F). Interestingly in both papers mentioned above, the authors suggest that these two layers of transcriptional control of the UPR might allow mammalian cells to deal with low level and chronic ER stress by activating only the ATF6 arm of the UPR. This arm is able to refold proteins and prevent further stress and activation of XBP1u as per the discussion in PMID: 12586069. Moreover as shown in another paper, PMID: 17765679, the presence of a functioning ATF6 allows cells to recover from acute and persistent stress and be preconditioned to secondary stress so that they do not commit to execute other branches of UPR or the UPR-dependent apoptotic programmes (see Figure 6 and S8B of PMID: 17765679). This is also confirmed in the previously mentioned PMID:21521793 where the authors demonstrate that ATF6 activation may serve a preemptive function in avoiding the problems that would otherwise be caused by membrane protein overload. Our ProteoStat data mentioned above (Extended Data Fig.5B) do show a significant, but only marginal increase in protein aggregates in MPI KO cells which might be consistent with a situation of mild persistent ER stress. Indeed we speculate that the chronic effects of MPI depletion, through activation of the ATF6 arm of the UPR, prevent ongoing activation of the other branches of the UPR response. This chronic activation and the previous observation that mammalian cells only rely on the IRE1 axis of the UPR for more acute and pronounced ER stress and once the refolding ability driven by ATF6 is overwhelmed (see discussion of PMID: 12586069 and PMID: 11779464) might reconcile our data with the mentioned literature. We have added a sentence regarding this in the results.

6) The use of pharmacological agents to modulate the ATF6 arm of the UPR should be i) confirmed in the cell system to prove that they either prevent or activate the cleavage of ATF6, ii) confirmed with other pharmacological agents (at least for the inhibition part) with for instance inhibition of ATF6 cleavage using serine protease inhibitors, and iii) also confirmed using genetic approaches (eg siRNA or overexpression of ATF6f).

i) We have now performed WB for ATF6 in MPI KO cells treated with Ceapin A7 (ATF6 inhibitor) and observe a clear reduction in the cleavage of ATF6 with Ceapin A7 (Extended Data Fig 5G). This is consistent with our immunofluorescence data for ATF6 following Ceapin A7 treatment which was presented in Fig 5C (now showed at improved resolution). Here we had shown that indeed Ceapin A7 reduces the amount of nuclear ATF6. Incidentally these data also provide validation of the WB bands observed in the ATF6 blots. With regards to AA147 (reported ATF6 activator) we have not detected a consistent increase of ATF6 cleavage by WB at the timepoints assessed. However by qPCR we observe that AA147 increases expression of ATF6 targets (Extended Data Fig5M) and here (left panel) we show that it appears to preferentially activate ATF6 targets rather than the PERK target ATF4 at 24hours post treatment. We now refer to AA147 in the manuscript as UPR activator with reported preferential activity on ATF6 ii) We have tried to confirm the effects of ATF6

inhibition using another commonly used ATF6 inhibitor, i.e. the serine protease 1 inhibitor PF-429242. However here we did not observe a rescue and in fact PF-429242 alone was actually toxic to our cells possibly because this inhibitor is known to block cleavage of other proteins by SP1 such as SREBP1 which might be essential for leukemia cell survival (see PMID: 34293334). It is worth noting that Ceapin A7 is considered to be a more specific inhibitor of ATF6 compared to SP1 inhibitors through its mode of action which is independent of SP1 inhibition (see Gallagher et al., eLife 2016 and references 35 and 36 in the manuscript). We have added a panel here on the right for the Reviewer showing the data using PF-429242 following 72 hours of treatment. iii) We silenced ATF6 in our cells by siRNA and as shown in Extended Data Fig.5H-I we can effectively phenocopy the effects of Ceapin A7.

7) Beyond ATF6, the other arms of the UPR were also shown to be linked to the control of ferroptosis activation and therefore should be documented in the authors' models.

As stated above in point 4, we do not observe a rescue in viability of our MPI KO cells treated with cytarabine or AC220 following the addition of either PERK or IRE1 inhibitors (Extended Data Fig.5L). Moreover we do not observe any changes in lipid peroxidation as measured by Bodipy staining in our cells in the same conditions thus suggesting that at least in our system these 2 branches of the UPR response are less relevant in modulating ferroptosis induction and cell viability (Extended Data Fig. 6H-I).

Reviewer #3 (Remarks to the Author):

Woodley et al. identify that mannose-6-phosphate isomerase (MPI) knockout sensitized therapy resistant acute myeloid leukemia (AML) cells to FLT3-tyrosine kinase inhibitor (TKI) AC220 (quizartinib) and AraC. They attempt to identify by which mechanism MPI knockout sensitizes to conventional therapy and propose that increased fatty acid uptake (via increased CD36 expression) and unfolded protein response (UPR) leads to activation of ATF6 and subsequent reduced mitochondrial fatty acid oxidation resulting in increased amounts of free PUFAs promoting ferroptotic cell death under treatment with AC220 and AraC.

While the authors identify an interesting connection of MPI in the mannose pathway and therapy resistance in AML, the extent to which this relates to ferroptosis is entirely unclear. The mechanism of how MPI knockout increases fatty acid uptake and reduces FAO is not sufficiently connected to the mechanism of ferroptosis and misses important control in vivo/in vitro experiments .

Major points:

In Figure 2F: it is unclear whether the phenotype has anything to do with ferroptosis. Given the direction of the paper, they need to rescue AraC/AC220 treatment with Liproxstatin-1 to show that the effect is mediated by ferroptotic cell death and include 4-HNE stains *in vivo*.

We thank the Reviewer for the proposed experiment which is needed to further support the role of ferroptosis in the observed phenotype *in vivo*. We performed this experiment using the THP1 cell line given that as per Figure 2F, we observed a longer latency following treatment with cytarabine in mice transplanted with MPI KO cells compared to those transplanted with control gRNA cells in this model compared to the MOLM13 model. To reduce the unnecessary use of mice, we only performed the experiment in the cytarabine treated cohorts. As shown in Fig. 6E, mice transplanted with THP1 MPI KO cells and treated with cytarabine showed longer disease latency than similarly treated mice transplanted with THP1 control gRNA cells consistent with data previously shown in Fig 2F. Intraperitoneal treatment of mice with liproxstatin-1 however reduced the survival of AraC treated mice transplanted with MPI KO cells thus suggesting that inhibition of ferroptosis promoted AML cell survival *in vivo* and led to reduced disease latency. Moreover we also analysed a cohort of mice soon after completion of therapy to assess lipid peroxidation changes in AML cells *in vivo* using 4-HNE staining by immunofluorescence. Although soon after therapy most mice had very limited disease burden, we were able to sort enough human cells from their marrow (Extended Data Fig. 7I) to perform 4-HNE staining by immunofluorescence. As shown in Fig. 6F and extended Data Fig. 7J we indeed observe an increased 4-HNE staining in MPI KO AraC treated cells *in vivo*. We therefore believe this experiment has provided a further validation *in vivo* of the ferroptotic cell death previously observed *in vitro* in MPI KO cells. We have also attached here for the Reviewer the 4-HNE immunofluorescence for all the mice culled soon after therapy delivery with pictures taken with different contrast enhancement to highlight the differences. In the “MPIgRNA5 + AraC” condition the normal and high contrast images look the same due to the method of contrast enhancement deployed by ImageJ. ImageJ uses a target percentage of saturated pixels in the enhanced contrast image (in this case 0.5%) so if the percentage of saturated pixels in the original image is at or above this threshold there will appear to be no difference between the original and high contrast images. However if there are fewer saturated pixels than the target in the original image the contrast will be enhanced.

The specificity of 4-HNE antibody staining was tested by treating WT THP-1 cells with 50 μ M tert-butyl hydroperoxide (TBHP) to induce 4-HNE formation or PBS as a control before mounting and staining for immunofluorescence. In this case we saw a significant increase in the fluorescence on the 4-HNE channel in the TBHP treated condition.

What type of cell death do therapy (AC220, AraC)-sensitive parental AML cells undergo upon treatment, is it ferroptosis or other types of cell death? The mechanism of AC220 and AraC mediated therapy is very different, why would they have the same mechanism of resistance? They need to perform rescue experiments with Fer-1, zVAD, Nec-1 to rescue different types of cell death

We thank the Reviewer for suggesting experiments aiming to further clarify the mechanism of cell death in our system. With regards to what type of cell death wild-type AML cells undergo following AraC or AC220 therapy, this has been previously shown in the literature as being mostly apoptotic (Gallipoli et al., Blood 2018; Kampa-Schittenhelm et al, Mol Cancer 2013; Li et al., Oncotarget 2016 PMID: 27462781). However it is also worth noting that at the concentration we used in our manuscript, both AraC or AC220 were causing mostly antiproliferative effects with some induction of cell death on their own which was enhanced in the MPI KO cells. With regards as to why these 2 different drugs would elicit the same mechanism of resistance, it is worth noting that reliance on OxPhos has been highlighted as a common feature of cells resistant to various therapy in AML (see also comments to point 3 of Reviewer 1). We and other have previously shown that FLT3-inhibitor resistant cells rely more strongly on OxPhos which can be fuelled by anaplerotic substrates such glutamine and fatty acids (Gallipoli et al., Blood 2018, Joshi et al. Cancer Cell 2018). Similarly work from several other groups has shown that both resistance to AraC and other novel therapies such as venetoclax can be driven by fatty acid oxidation (FAO) driven OxPhos (Farge et al., Cancer Discovery 2017; Bosc et al., Nature Cancer 2021; Stevens et al., Nature Cancer 2020). It is therefore not surprising that resistance to both AraC and FLT3 inhibitors converge on enhanced OxPhos fuelled also by FAO. MPI KO by driving a defective FAO via its effects on the ATF6 arm of the UPR response (as shown in Fig 5) therefore sensitizes to both therapies by removing a key mechanism driving resistance in both instances. Moreover through its effects on PUFAs accumulation (caused by both defective FAO and increased uptake) it also leads to a status primed for ferroptosis which then becomes the common effector mechanism of cell death in both MPI KO cells treated with AraC and AC220. To further support this hypothesis, we have performed as suggested viability rescue experiments with zVAD and Nec-1 and could not elicit any significant rescue

in viability in our cells (Extended Data Fig. 7H) thus further supporting that ferroptosis is the main driver of cell death in treated MPI KO cells.

Figure 6 is the only figure attempting to link the mechanism to ferroptosis. However, given that this is a prominent novelty of the manuscript, they need to include these validations in all mechanistic corner points of the manuscript, e.g. is the MPI ko lipid ROS phenotype rescued by ATF6/CD36/etc. inhibition? Can they rescue lipid ROS induction in MPI ko with Fer-1? What about the response of MPI ko cells to all commonly used ferroptosis inducers (RSL2, ML210. Etc.).

We agree that in order to strengthen the link of cell death to ferroptosis, these experiments are required. We have now shown that MPI KO cells are more sensitive to both erastin and RSL3 two ferroptosis inducing agents acting on 2 key proteins regulating ferroptosis, i.e. SLC7A11 and GPX4 (Extended Data Fig. 7E). We have also looked at the effects of inhibiting all the different hubs in the proposed mechanistic cascade on the lipid ROS phenotype and have observed that indeed Ferrostatin as expected rescues lipid peroxidation (Extended Data Fig. 6E). Interestingly ATF6 inhibition by Ceapin A7 also does rescue lipid peroxidation while ATF6 activation induces lipid peroxidation in MPI KO cells after mannose rescue (Extended Data Fig. 6F-G). CD36 inhibition however does not rescue lipid peroxidation and we speculate this might be due to the fact that it affects uptake of only some fatty acid species (Extended Data Fig. 3G-H) and has no effects on FAO. As a result CD36 inhibition is blocking only some of the effects of MPI KO (uptake of PUFAs but not defective FAO). Given that defective FAO can on its own lead to PUFAs accumulation by reduced oxidation, CD36 inhibition might not be enough to reverse the lipid ROS production (Extended Data Fig. 7D). Moreover it is also possible that PUFAs uptake could be mediated via other transporters not blocked by SSO thus limiting its ability on its own to block the lipid ROS formation. These data also suggest that sensitivity to arachidonic acid toxicity, which we report in Extended Data Fig. 7A-B for both Control and MPI KO cells, is also not just due to increased lipid peroxidation. Therefore we have added a sentence in the results on the effects of arachidonic acid being not just secondary to induction of lipid peroxidation. However it is worth noting that in these latter lipid ROS experiments (Extended Data Fig. 7D) we did not feed cells exogenous arachidonic acid in excess as per Extended Data Fig. 7A-B.

They see a difference in total lipid uptake, what about actual lipid peroxidation in MPI ko cells?

We do indeed see a difference in lipid peroxidation between MPI KO cells using the lipid ROS detection staining with bodipy C11 as per figure 6C. This is however only seen upon treatment of cells with AC220 or cytarabine possibly because lipid peroxidation needs both lipid accumulation and oxidative stress triggering from the therapies.

Which of the many things regulated in MPI ko cells is responsible for ferroptosis sensitization? Cd36 upregulation, xCT regulation, cystine uptake, differences in GSH levels?

As per the point above, we do see that MPI KO cells are generally more primed to ferroptosis and that blocking any of the hubs below MPI KO (i.e. ATF6 activation) can rescue the ferroptotic cell death possibly because it will revert all the effects of MPI KO on both FAO and lipid uptake. On the other end blocking only CD36 and lipid uptake is not able on its own

to revert lipid peroxidation. We believe this is due to the reasons outlined in the above point.

Minor points:

Figure 1:

- A: labels need to be bigger – As this panel was also duplicated in Extended Fig. 1A, it has been removed from Figure 1A. The labels in Extended Fig. 1A have been changed
- F: not consistent with color scheme (MPI high red) – This has been changed in what is now Fig 1E

Figure 3:

- A: Why does mannose supplementation increase the amount of PUFA in WT cells?

We agree with the Reviewer that this was a somewhat puzzling aspect of our metabolic analysis. However we note that the increase is significantly less than in MPI KO treated cells so likely less significant functionally. We also speculate that mannose in WT cells is effectively channelled towards glucose metabolism by the still active MPI thus limiting the utilisation of uptaken PUFAs in WT cells treated with AC220. This might in turn lead to a small increase in their intracellular levels.

- C: please show different FACS plot (half offset) for better overview. Should there not be an increase with AC220 which is then rescued with mannose?

We have now shown half offset FACS plots. Also the bodipy 500/510 probe for lipid uptake is looking at different time point compared to Figure 3A (the incubation with bodipy was for 24 hours versus 48 hours in Fig 3A) and also it is possible that by then the AC220 treated MPI KO cells were not fully reliant on FAO hence lipid uptaken was not metabolised in the presence of mannose hence levels remaining still high

- A and D do not show the same results, what is the normalization in A?

Indeed as mentioned above these 2 experiments show different assays at different timepoints. For Figure 3D intact cells were also loaded for 24 hours with fatty acids which would make the comparison very difficult. The data shown in Figure 3A are normalised by Bradford protein concentration and then each biochemical is rescaled to the median levels of each biochemical across all 6 conditions which is set as reference. Z scores are then calculated as the difference between the observed value in one condition minus the mean, all divided by the standard deviation, i.e. $(x-\mu)/\sigma$. This explanation has been added to the methods.

Ext. Figure 6:

- A/B: How do you explain that ac220 does not sensitize more to ferroptosis but rescue from it, like mannose and CD36 inhibition?

Note this is now Extended Fig. 7A-B. We note that the difference/rescue in cell kill with AC220, which is only seen in MPI KO cells, is actually not significant. It is also noticeable that arachidonic acid causes a dramatic induction of cell death in MPI KO cells upon which further sensitization is impossible. Also in MPI WT cells we also observe reduction of

viability to below 60% in cells treated with arachidonic acid suggesting that the mechanism of cell death is likely multifactorial and not just ferroptotic (see also the lack of rescue in peroxidation with SSO) and that the interaction with AC220 is more complex. Interestingly in Fig. 3D AC220 also trends towards a reduction of arachidonic acid uptake without providing the rescue in uptake given by mannose or SSO. This might also explain the trend observed in viability. However we believe that the effects of exogenously supplemented arachidonic acid on the cells are likely multifactorial and therefore interactions with AC220 might be complex and more nuanced than the effects of mannose or CD36 inhibition which by reducing the uptake of arachidonic acid/other fatty acid species (the latter at least for mannose) in MPI KO treated cells provide a rescue at the source for all the downstream effects of arachidonic and other fatty acids.

Figure 7:

- C: misleading description (not clear whether bodipy c11)

This has now been changed

REVIEWERS' COMMENTS

Reviewer #1 (Remarks to the Author):

The authors have thoughtfully and thoroughly addressed all of my comments.

Reviewer #2 (Remarks to the Author):

The authors did a good job at addressing this reviewer's comments, there is no further experimental work required.

Reviewer #3 (Remarks to the Author):

All major points raised have been addressed by the authors .

Minor point:

For rescue experiments checking for necroptosis induction Nec-1s should be used instead of Nec-1 as it rescues ferroptotic cell death as well in a RIPK1-independent manner. So the authors should consider commenting on this in the manuscript

We would like to thank the Reviewers for their positive feedback on our revised manuscript and also for their previous comments and requests, all of which have improved our manuscript.

In response to Reviewer 3 minor point, we have added a sentence regarding the specificity of Necrostatin-1 (Nec-1) in our results section on page 9 and added a reference (47) to highlight that Nec-1 can indeed rescue ferroptosis in other cellular models although we did not observe this in our system. As we observe ferroptosis rescue with canonical ferroptosis inhibitors and given Nec-1 known action as necroptosis inhibitor, we interpret that its lack of efficacy is due to the fact that necroptosis is not a driver of cell death in our system.

REVIEWERS' COMMENTS

Reviewer #1 (Remarks to the Author):

The authors have thoughtfully and thoroughly addressed all of my comments.

Reviewer #2 (Remarks to the Author):

The authors did a good job at addressing this reviewer's comments, there is no further experimental work required.

Reviewer #3 (Remarks to the Author):

All major points raised have been addressed by the authors .

Minor point:

For rescue experiments checking for necroptosis induction Nec-1s should be used instead of Nec-1 as it rescues ferroptotic cell death as well in a RIPK1-independent manner. So the authors should consider commenting on this in the manuscript